

# Molecular distribution and compound-specific stable carbon isotopic composition of dicarboxylic acids, oxocarboxylic acids, and α-dicarbonyls in PM$_{2.5}$ from Beijing, China

Wanyu Zhao[1,2,3], Kimitaka Kawamura[2,4], Siyao Yue[1,5], Lianfang Wei[1,5], Hong Ren[1,5], Yu Yan[1], Mingjie Kang[1], Linjie Li[1,5], Lujie Ren[6], Senchao Lai[7], Jie Li[1], Yele Sun[1], Zifa Wang[1], and Pingqing Fu[1,5,6]

[1] State Key Laboratory of Atmospheric Boundary Layer Physics and Atmospheric Chemistry, Institute of Atmospheric Physics, Chinese Academy of Sciences, Beijing 100029, China

[2] Institute of Low Temperature Science, Hokkaido University, Sapporo 060-0819, Japan

[3] Chengdu University of Information Technology, Chengdu 610225, China

[4] Chubu Institute for Advanced Studies, Chubu University, Kasugai 487-8501, Japan

[5] College of Earth Sciences, University of Chinese Academy of Sciences, Beijing 100049, China

[6] Institute of Surface-Earth System Science, Tianjin University, Tianjin 300072, China

[7] Guangdong Provincial Engineering and Technology Research Center for Environmental Risk Prevention and Emergency Disposal, School of Environment and Energy, South China University of Technology, Guangzhou, China

Correspondence to: Pingqing Fu (fupingqing@mail.iap.ac.cn)



## Abstract

This study investigates the seasonal variation, molecular distribution and stable carbon isotopic composition of diacids, oxocarboxylic acids and α-dicarbonyls to better understand the sources and formation processes of fine aerosols ($PM_{2.5}$) in Beijing. The concentrations of total dicarboxylic acids varied from 110 to 2580 ng m$^{-3}$, whereas oxoacids (9.50–353 ng m$^{-3}$) and dicarbonyls (1.50–85.9 ng m$^{-3}$) were less abundant. Oxalic acid was found to be the most abundant individual species, followed by succinic acid or occasionally by terephthalic acid (tPh), a plastic burning tracer. Ambient concentrations of phthalic acid (37.9±27.3 ng m$^{-3}$) and tPh (48.7±51.1 ng m$^{-3}$) are larger in winter than in other seasons, illustrating that fossil fuel combustion and plastic waste incineration contribute more significantly to wintertime aerosols. The year-round mass concentration ratios of malonic acid to succinic acid ($C_3/C_4$) are relatively by comparison with those in other urban aerosols and remote marine aerosols, most of which are less than or equal to unity in Beijing; thus, the degree of photochemical formation of diacids in Beijing is insignificant. Moreover, positive correlations of some oxocarboxylic acids and α-dicarbonyls with nss-$K^+$, a tracer for biomass burning, suggest biogenic combustion activities accounting for a large contribution of these organic acids and related precursors. The mean $\delta^{13}C$ value of succinic acid is highest among all species with values of −17.1±3.9‰ (winter) and −17.1±2.0‰ (spring), while malonic acid is less enriched in $^{13}C$ than others in autumn (−17.6±4.6‰) and summer (−18.7±4.0‰). The $\delta^{13}C$ values of major species in the Beijing aerosols are generally lower with a wider range than those in downwind regions in the western North Pacific, which indicates that Beijing has diverse emission sources with weak photooxidation. Thus, our study demonstrates that in addition to photochemical oxidation, high abundances of diacids, oxocarboxylic acids and α-dicarbonyls in Beijing are largely associated with anthropogenic primary emissions, such as biomass burning, fossil fuel combustion, and plastic burning.



# 1 Introduction

Haze pollution events are largely characterized by high levels of fine aerosol particles ($PM_{2.5}$) and have received considerable public attention in China during the past few years (Cao, 2012; Zhang et al., 2014; Zhao et al., 2013). $PM_{2.5}$ influences air quality, visibility, human health, radiative forcing and global climates (Ulbrich et al., 2009; Sun et al., 2013) and is heavily involved with organic aerosols, making up 20–50% of aerosol mass (Kanakidou et al., 2005) and no less than 90% in tropical forest areas (Falkovich et al., 2005). Interestingly, large quantities of organic aerosols are water-soluble, resulting in corresponding proportion from 20% to 75% of total carbon mass in particles, which originate from incomplete combustion activities (biomass burning: 45–75%, fossil fuel burning: 20–60%) (Pathak et al., 2011; Falkovich et al., 2005). Due to their hygroscopic properties, water-soluble organic aerosols (WSOA) act as an important role in global climate change by influencing solar radiation (Facchini et al., 1999; Saxena et al., 1995).

Homologues series of diacids, oxoacids and α-dicarbonyls comprise a major portion of WSOA (Kawamura and Ikushima, 1993; Miyazaki et al., 2009). Owing to the existence of two carboxyl groups, diacids are less volatile and highly water soluble, and they play an important role in acting as CCN to affect the earth's radiative balance. (Kanakidou et al., 2005; Andreae and Rosenfeld, 2008). They are widely present in urban (Ho et al., 2007), rural (Kundu et al., 2010a), marine (Fu et al., 2013), and the Arctic atmospheres (Kawamura et al., 1996a). Concentrations of total diacids contribute approximately 1–3% to the total carbon mass in urban regions and more than 10% in remote marine atmospheres (Kawamura and Ikushima, 1993; Kawamura et al., 1996b; Kawamura et al., 1996c). Diacids, ketoacids and α-dicarbonyls not only can be directly released from primary emissions like biomass burning (Turnhouse, 1987; Destevou et al., 1998; Schauer et al., 2001; Kundu et al., 2010a), meal cooking (Rogge et al., 1991; Schauer et al., 1999; Zhao et al., 2007), fossil fuel burning (Kawamura and Kaplan, 1987; Rogge et al., 1993) along with motor vehicles (Kawamura and Kaplan, 1987; Donnelly et al., 1988), but also are largely produced by photooxidation reactions during atmospheric transport (Kawamura and Yasui, 2005; Kundu et al., 2010b).



Breakdown of relatively long carbon-chain diacids and other related precursors is also one of
the key sources of low carbon-numbered diacids in the atmosphere (Agarwal et al., 2010).
Realizing the physical and chemical characteristics of organic matters is vital for determining
the source regions and elucidating the mechanism of evolution of air pollution events.
Various measurements have been employed for closer acquaintances of the sources,
transformation and long-distance transport of organic compounds, including studies on sugars,
unsaturated fatty acids, *n*-alkanes and *n*-alcohol, along with aromatic hydrocarbons
(Kawamura and Gagosian, 1987; Kawamura et al., 1996a). Zhang et al. (2010) conducted
field observations of dicarboxylic acids and pinene oxidation products with a model analysis
of the temperature dependencies of emissions, gas-particle partitioning, and chemical
reactions. Furthermore, the analyses of stable carbon isotope ratios of water-soluble organic
acids can be effectively applied to assessing the photochemical aging of aerosol samples in
atmosphere (Kawamura and Watanabe, 2004; Wang and Kawamura, 2006c). With this
approach, it is possible to differentiate between the impacts of local sources and long-range
transported air masses.
Beijing, the capital of China, is located on the northwest rim of the North China Plain and is
embraced with industrialized areas from the southwest to the east. Emissions from local and
regional sources potentially undergo photochemical processes in the course of transport by
means of prevalent winds, which influence the atmospheric visibility and quality in Beijing
(Xia et al., 2007). Several studies have reported that the source strength of aerosols in Beijing
is characterized by fossil fuel combustion in winter, whereas it is characterized by secondary
aerosol formation in summer (Lin et al., 2009; Sun et al., 2015; Wang et al., 2006). Ji et al.
(2016) observed increasing photochemical activity in autumn and winter and noted biomass
burning as a substantial pollution factor in Beijing. In addition to the studies on long-term
observations of organic aerosols, specific haze pollution episodes occurring in Beijing have
been investigated. For example, Huang et al. (2014) concluded that in comparison with
secondary sources, primary emissions contributed slightly less to fine particles in haze events
at urban locations in China, including Beijing, using both molecular markers and radiocarbon
($^{14}$C) measurements. To ascertain the influential factors for air quality in Beijing, two studies





demonstrated that besides vehicle emissions, oxidation pathways of organic species is also
critical (Ho et al., 2010; Ho et al., 2015). Although such studies have focused on the
characterization of organic aerosols in Beijing at a molecular level, long-term analyses of low
molecular weight (LMW) dicarboxylic acids, oxoacids and α-dicarbonyls with their stable
carbon isotopic compositions have not been investigated.
To better understand the sources, photochemical processes, and seasonal distributions of
organic aerosols in Beijing, $PM_{2.5}$ samples were collected from September 2013 to July 2014.
These samples were analyzed for organic carbon (OC), elemental carbon (EC), water-soluble
organic carbon (WSOC) and inorganic ions. In addition to reporting the concentrations and
seasonal variations of LMW dicarboxylic acids, oxoacids and α-dicarbonyls, we investigated
the seasonal trends of stable carbon isotopic compositions of these water-soluble organic
acids. Using these measurements, the contribution of primary emissions, long-range transport,
as well as photochemical production ways of organic matters in Beijing were examined. The
effects of air masses on aerosol compositions and formation mechanisms are also discussed.

## 123  2  Experimental section

### 124  2.1  $PM_{2.5}$ sampling

The sampling site is situated on the rooftop of a building (8 meters above ground level) in the
Institute of Atmospheric Physics (39°58'28''N, 116°22'16''E), which is considered as a
representative urban site in Beijing (Sun et al., 2012). $PM_{2.5}$ samples were collected onto
pre-heated (450°C, 6 hours) quartz-fiber filters (Pallflex) by using a high-volume air sampler
(TISCH, USA) at an airflow rate of 1.0 $m^3$/min from September 2013 to July 2014 (n=65).
Field blanks were prepared before, during and after the campaign by putting the
pre-combusted filter onto the sampler for a few minutes without pumping. Beijing is
surrounded by Hebei Province and Tianjin Municipality with intensely developed industries
(Xia et al., 2007), so the atmospheric visibility and quality in Beijing are sometimes seriously
deteriorated owing to the substantial primary aerosols from these areas.



## 2.2 Analytical procedures

Aerosol samples were analyzed for diacids and related compounds using a method reported previously (Kawamura, 1993; Kawamura and Ikushima, 1993). In brief, water-soluble organic acids were obtained from ultrasonic extraction for small discs of $PM_{2.5}$ samples merged in Milli-Q water for 3 times. Then, the sample extractions were concentrated into dryness and further reacted with 14% $BF_3$/$n$-butanol. Finally, the derivatized extracts were dissolved in $n$-hexane and analyzed by a split/splitless Agilent 6980 GC/FID equipped with an HP-5 column (0.2 mm × 25 m, 0.5 μm film thickness). The field blank filters were also used same procedures to analyze. Concentrations of the target organic acids in this study were corrected for the field blanks. Furthermore, recoveries of major organic acids of this method were better than 85%.

## 2.3 Measurement of isotopic compounds

Determination of stable carbon isotope ratio ($\delta^{13}C$) values for LMW diacids and related constituents relative to Pee Dee Belemnite (PDB) were used the technique set up previously (Kawamura and Watanabe, 2004). In short, $\delta^{13}C$ values of the derivatized dibutyl esters or dibutoxy acetals, measured using GC (HP6890)/isotope ratio mass spectrometer (irMS), were calculated for diacids, ketoacids and α-dicarbonyls. Every aerosol sample was analyzed for several times to make sure that the differences for major diacids in $\delta^{13}C$ below 1‰ in general. But as to few compounds, the analysis differences were less than 2‰.

## 2.4 Inorganic ions, WSOC, OC and EC measurements

A part of each filter was extracted with 20 ml of Milli-Q water under ultrasonication for 30 minutes and passed through a filter head of 0.22 μm nominal pore size (PVDF, Merck Millipore Ltd). Ion chromatography (ICS-2100) was used to determine the concentrations of cations ($Na^+$, $NH_4^+$, $K^+$, $Mg^{2+}$, and $Ca^{2+}$). The separation of cations was accomplished by using an IonPac CS 12A (4×250 mm) analytical column, with an eluent flow rate of 1.0 ml/min. Another ICS-2100 system was used to measure the concentrations of anions ($F^-$, $Cl^-$, $NO_3^-$, and $SO_4^{2-}$). The separation of anions was accomplished using an IonPac AS11-HC analytical column. The eluent was 25.0 mM KOH at a flow rate of 1.0 ml/min. The anions





and cations were analyzed separately after the extraction solution was divided into two paths.
For the WSOC measurement, 3.14 cm$^2$ of each filter was extracted by Milli-Q water (20 ml).
After 15 min sonication, the extraction was measured by Shimadzu TOC-V CPH Total
Carbon Analyzer (Aggarwal and Kawamura, 2008). OC and EC were determined by using
thermal optical reflectance (TOR) following the Interagency Monitoring of Protected Visual
Environments (IMPROVE) protocol on a DRI Model 2001 Thermal/Optical Carbon Analyzer
(Chow et al., 2005). The limit of detection (LOD) for the carbon analysis was 0.8 μgC cm$^{-2}$
for OC and 0.4 μgC cm$^{-2}$ for EC, with a precision of greater than 10% for total carbon (TC).
The concentrations of inorganic ions, WSOC and OC/EC reported here are all corrected for
the field blanks.

**2.5   Air mass backward trajectories**

To better evaluate the influences of air masses from different origins on organic aerosols in
Beijing, 5-day backward trajectory analyses with fire spots were performed for each sample
from the sampling site at a height of 500 m (a.s.l.) by using the Hybrid Single-Particle
Lagrangian Integrated Trajectory (HYSPLIT4) mode (Rolph, 2011, 2003). And fire spot data
were           downloaded           from           the           MODIS           website
(https://firms.modaps.eosdis.nasa.gov/download/request.php).   The   backward   trajectories
were assorted into several major classifications on the basis of prevalent winds direction, as
given below in this study (Fig. 1).

# 3   Results and discussion

## 3.1   Molecular distribution

Table 1 summarizes the seasonal concentrations of LMW diacids ($C_2$–$C_{12}$), oxocarboxylic
acids ($\omega C_2$–$\omega C_9$, pyruvic acid), and α-dicarbonyls ($C_2$–$C_3$) in $PM_{2.5}$ particles in Beijing.
Oxalic acid (30.8–1760 ng m$^{-3}$, average 288 ng m$^{-3}$) was the predominant individual diacid,
showing a peak in autumn and a minimum in winter, whereas its relative abundances (0.39–
0.58, average 0.52) to total measured species exhibited a maximal and a minimal ratio in





summer and wintertime, respectively (Table 3). The predominance of $C_2$ found in this study
was coincident with the results of terrestrial and marine particles in previous studies
(Kawamura and Yasui, 2005; Ho et al., 2007; Pavuluri et al., 2010; Fu et al., 2013). Among
ω-oxocarboxylic acids (ω$C_2$–ω$C_9$), glyoxylic acid (ω$C_2$) was detected as the dominant
oxoacid.
Either succinic acid ($C_4$) or occasionally tPh was the second most abundant compound,
followed by ω$C_2$ in cold seasons (autumn and winter) or malonic acid ($C_3$) in warm seasons
(spring and summer). Total diacids showed the largest abundance in autumn, followed by
spring, whereas total oxoacids and α-dicarbonyls both displayed higher levels in cold seasons,
especially in autumn (Table 1). The concentrations of single dicarboxylic acid reduced along
with the increase of carbon numbers, but in the range of longer-chain diacids ($C_6$–$C_{12}$), adipic
($C_6$) and azelaic ($C_9$) acids showed more abundances than other species in the atmosphere
throughout the year (Fig. 2).
**3.2   Seasonal variations**
Seasonal trends of homologues series of diacids and other main species presented three
different patterns. The first type, such as oxalic, malonic, succinic, adipic acids and
methylglyoxal (MeGly), showed maximum concentrations in autumn with relatively high
abundances in late spring to early summer. The second type showed maximum concentrations
in cold seasons (autumn, winter): phthalic, terephthalic and glyoxylic acid peaked in winter,
while $C_9$, glyoxal (Gly), methylmaleic (mM), maleic (M) and fumaric (F) acids peaked in
autumn. For the third type, concentrations of methylmalonic (i$C_4$) and 2-methylglutaric (i$C_6$)
acids were almost constant throughout the sampling year. These three seasonal trends
indicated different emission sources of the compounds and their precursors and evolution
processes of organic aerosols in the atmosphere.
Total concentrations of diacids showed a wide range (110–2580 ng m$^{-3}$) with an average
maximum (763 ng m$^{-3}$) and minimum (366 ng m$^{-3}$) in autumn and winter, respectively. These
values were comparable to those in Tanzania, East Africa (wet season: 329 ng m$^{-3}$; dry
season: 548 ng m$^{-3}$ in PM$_{2.5}$) (Mkoma and Kawamura, 2013), slightly lower than those in



Tokyo, Japan (726 ng m$^{-3}$ in June, 682 ng m$^{-3}$ in July, and 438 ng m$^{-3}$ in November)
(Kawamura and Yasui, 2005), and Gosan, Jeju Island in Korea (735 ng m$^{-3}$ in spring, 784 ng
m$^{-3}$ in summer, 525 ng m$^{-3}$ in autumn, and 500 ng m$^{-3}$ in winter) (Kundu et al., 2010a). The
comparisons of the diacids in Beijing with those in other urban cities are presented in Table

222    2.

Daily variations of diacids and other major organic acids are given in Figure 3. Oxalic acid
has been recognized as the end product that is associated with atmospheric chain reactions of
organic species with oxidants (Kawamura and Sakaguchi, 1999). $C_2$ can be generated in
abundant quantities by vehicular emissions (Kawamura and Kaplan, 1987; Donnelly et al.,
1988), biomass burning activities (Turnhouse, 1987; Destevou et al., 1998; Schauer et al.,
2001), fossil fuel combustion (Rogge et al., 1993), and photo-oxidation of volatile organic
compounds and other precursors transported from long distance (Kawamura and Yasui, 2005;
Kundu et al., 2010b). Malonic acid was detected at relatively low concentrations in four
sampling periods, with the highest abundances in autumn. The concentrations of $C_4$ diacid in
excess of $C_3$ diacid implies that primary emissions contributed more to dicarboxylic acids, a
typical pattern that is frequently obtained in aerosols emitted from biomass burning
(Kawamura et al., 2013), vehicular exhaust (Ho et al., 2010) and fossil fuel combustion (Ho
et al., 2007). The diurnal variation tendency of $C_2$ resembled to that of both $C_3$ and $C_4$, with
the trends $C_3$ and $C_4$ being almost the same, indicating that these compounds may have
similar photochemical oxidation pathways or emission sources in the atmosphere.
In addition to shorter-chain diacids ($C_2$–$C_4$), azelaic acid ($C_9$) had the highest concentration
among the saturated diacids in all seasons (Table 1). $C_9$, a photochemical oxidation product
of unsaturated fatty acids derived from natural biogenic sources such as terrestrial higher
plants and sea-to-air emission of marine organics, as well as anthropogenic emissions
including biomass burnings (Kawamura and Gagosian, 1987). Under favorable atmospheric
conditions, photooxidation of biogenic unsaturated fatty acids to $C_9$ with oxidants, such as $O_3$,
OH and $HO_2$, are inclined to occur in air (Stephanou and Stratigakis, 1993). Additionally, tire
wear debris and traffic exhaust also make contributions to the abundances of LMW fatty
acids like $C_{18:0}$ in atmosphere (Rogge et al., 1993).



Azelaic acid was observed in abundance throughout the whole sampling period, while the
monthly mean ratios of $C_9$ to total diacids ($C_9$/Tot) ranged from 0.05 to 0.09, with the highest
values in winter (Table 3). Kawamura and Kaplan (1987) reported that $C_9$ can be detected in
motor exhaust and may originate from the oxidation of corresponding hydrocarbons,
suggesting that dicarboxylic acids are combustion products of normal alkanes in fuels. A
great deal of chloride in wintertime Beijing is linked to increased emissions of coal
incineration, particularly under stagnant meteorological conditions that facilitate the
formation of particle-phase ammonium chloride (Sun et al., 2013). Azelaic acid correlated
well with $K^+$ ($0.3 \leq r^2 \leq 0.4$) and $Cl^-$ ($0.4 \leq r^2 \leq 0.5$) in cold seasons (Fig. S1), indicating that
substantial amounts of $C_9$ may be stemmed from the local and surrounding combustion
activities in Beijing.
Abundances of both Ph and tPh are higher in cold seasons than in warm seasons. We found
Ph to be the fourth most abundant species in winter ($37.9\pm27.2$ ng m$^{-3}$) and summer
($24.9\pm8.2$ ng m$^{-3}$). Concentration ranges of Ph (7.6–98.5 ng m$^{-3}$, mean: 31.7 ng m$^{-3}$) in cold
seasons were larger than those (0.08–7.47 ng m$^{-3}$, mean: 1.76 ng m$^{-3}$) from Gosan, Jeju Island
(Kundu et al., 2010b), but were obviously lower than these (53–278 ng m$^{-3}$, mean: 150 ng m$^{-3}$
) in urban Xi'an (Cheng et al., 2013). Phthalic acid is mainly formed via photochemical
pathways of naphthalene, and can also be directly released into air by fossil fuel burning and
the incomplete combustion of aromatic hydrocarbons in automobile emissions. A great
amount of naphthalene obtained in Beijing is an important raw material for the substantial
formation of phthalic acid (Liu et al., 2007). Thus, high concentrations of Ph in PM$_{2.5}$
demonstrate that vehicle emissions are one of the major pollution sources in Beijing.
Terephthalic acid (tPh), the second highest abundant diacid in winter ($48.7\pm41.1$ ng m$^{-3}$),
showed a pattern in contrast to a previous study that reported Ph as the second most abundant
compound (Ho et al., 2010). Terephthalic acid is directly emitted from plastic wastes
incinerations in ambient air (Simoneit et al., 2005; Kawamura and Pavuluri, 2010). High
concentration peaks of tPh observed in winter indicate substantial plastic waste incineration.
Another phthalic isomer, isophthalic acid (iPh), was also detected in the samples;
concentrations of this isomer had seasonal patterns similar to those of Ph and tPh throughout



the year. However, the concentrations of iPh were the lowest among the isomers.
Oxocarboxylic acids, which are understood as the intermediate products of the oxidation of
mono-carboxylic acids, can further be photo-chemically oxidized to form diacids.
Concentrations of all ketoacids varied from 9.50 to 353 ng m$^{-3}$ during sampling periods with
a maximum (73.3±76.3 ng m$^{-3}$) in autumn and a minimum (46.5±26.8 ng m$^{-3}$) in summer.
More abundant oxoacids, other than $\omega C_7$ and $\omega C_8$, in cold seasons (autumn and winter) might
be attributed to accumulation under stagnant meteorological conditions. Their concentrations
were higher than those in aerosols from Tanzania (60.0±19.0 ng m$^{-3}$ and 31.0±18.0 ng m$^{-3}$ in
PM$_{2.5}$ during dry and wet seasons, respectively) (Mkoma and Kawamura, 2013) but much
lower than these detected in Mangshan, a rural site in Beijing (159 ng m$^{-3}$ in daytime, 97.9 ng
m$^{-3}$ in nighttime) (He et al., 2014).
Glyoxylic acid ($\omega C_2$) is measured as the most abundant oxoacid, followed by pyruvic (Pyr)
and 4-oxobutanoic ($\omega C_4$) acids. All of them are important intermediates in photooxidation
processes for the production of low carbon-numbered diacids such as $C_2$, $C_3$ and $C_4$ diacids
(Hatakeyama et al., 1987). $\omega C_2$ and Pyr are more abundant in cold seasons (Table 1) with
similar seasonal patterns (Fig. 3 g–h) and have good correlations with K$^+$ (Fig. S2) in the
whole sampling period. $\omega C_2$ and Pyr only correlated well with Cl$^-$ (Fig. S3) in cold seasons.
These connections demonstrate that $\omega C_2$ and Pyr originated from common combustion
emissions or similar secondary formation pathways. 9-Oxononanoic acid ($\omega C_9$),
photochemically generated from unsaturated fatty acids (Kawamura and Gagosian, 1987),
showed larger concentrations in autumn and winter. This concentration trend was consistent
with that of azelaic acid. Additionally, a lower thermal inversion layer, less precipitation and
a slower wind speed can enhance the accumulation of organic compounds.
Total concentrations of α-dicarbonyls varied with a wide range (1.50–85.9 ng m$^{-3}$) and were
relatively more abundant in cold seasons (25.1±28.1 ng m$^{-3}$ in autumn, 15.5±15.9 ng m$^{-3}$ in
winter). And the average seasonal concentrations are larger than those at Gosan, Jeju island in
South Korea (Kundu et al., 2010b). Two α-dicarbonyls (glyoxal and methylglyoxal) are
semi-volatile gaseous organic precursors produced by oxidation of isoprene (Zimmermann
and Poppe, 1996), monoterpenes (Fick et al., 2004) and other biogenic volatile organic



compounds (VOCs) (Ervens et al., 2004) and anthropogenic aromatic hydrocarbons (e.g.,
benzene and toluene) (Volkamer et al., 2001). Both carbonyls can form less volatile organic
polar acids including Pyr and $\omega C_2$ in subsequent oxidation processes, which are key
intermediates to produce oxalic acid $(C_2)$. Glyoxal (Gly) and methylglyoxal (MeGly)
correlated well with nss-$K^+$ (Gly: $0.3 \leq r^2 \leq 0.9$, MeGly: $0.3 \leq r^2 \leq 0.9$) throughout the whole
year (Fig. S2), whereas Gly and MeGly showed good relations with $Cl^-$ (Gly: $0.3 \leq r^2 \leq 0.8$,
$0.4 \leq r^2 \leq 0.8$) in autumn, winter and summer (Fig. S3). Concentrations of these two
carbonyls are largely affected by biogenic precursors (e.g., isoprene and monoterpenes)
emitted from vegetation and biomass burning activities during entire sampling periods in
addition to coal burning and motor exhaust (aromatic hydrocarbons). Low temperature is
favorable for the adsorption and condensation of gaseous organic species on existing particles
in cold seasons.

### 3.3    Correlation analysis and seasonal variations of concentration ratios

The ratio of oxalic acid to total diacids $(C_2/Tot)$ has been applied to estimate the relative
contribution of secondary fraction to atmospheric aerosols during long-range transport.
Typically, higher mass concentration ratios are associated with more aged aerosols
(Kawamura and Sakaguchi, 1999). As for water-soluble organic acids in this paper, the ratios
of $C_2/Tot$ were the lowest in winter (0.39±0.05) (Table 3), indicating that wintertime organic
aerosols may be less aged (Fig. 4a). Because $PM_{2.5}$ particles mainly originate from motor
vehicles, fossil fuel and biomass combustion activities from local regions in winter, the aging
process might occur during atmospheric transport. In contrast, $C_2/Tot$ ratios are similar in the
other three seasons (Table 3). In total, the seasonal mean values of $C_2/Tot$ in this study were
lower than those in Central Himalayan in winter (0.8±0.04) owing to the aging of organic
compounds occurring in the northerly wind and were close to those in summer (0.5±0.01) due
to increased temperature and high wind in the Central Himalayas (Hegde and Kawamura,
2012). Thus, the ratios of $C_2/Tot$ and this seasonal trend indicate that the photochemical
formation of dicarboxylic acids is insignificant in urban Beijing.
Two pathways for the generation of $C_2$, $C_3$ and $C_4$ diacids in air were reported by Kawamura





et al. (1996a). On the one hand, $C_4$ diacid can be generated via the photooxidation of
unsaturated fatty acids from terrestrial higher plants and domestic cooking over continental
lands, as well as from phytoplankton emissions over the remote marine regions (Kawamura
and Sakaguchi, 1999), and subsequently be oxidized to form $C_3$ and $C_2$ diacids (Kawamura
and Ikushima, 1993). Typically, $C_2$/Tot ($C_2$%) showed strong correlations with $C_2/C_4$ in all
four seasons (Fig. S4), indicating the significance of photooxidation pathways of biogenic
unsaturated fatty acids. On the other hand, aromatic hydrocarbons may be oxidized to
produce Gly and $\omega C_2$, which are intermediates in the formation of $C_2$ (Kawamura and
Ikushima, 1993). Biogenic and anthropogenic VOCs (e.g., isoprene) can react with oxidants
to generate Gly and MeGly in the gas phase. Hydrated α-dicarbonyls can ultimately produce
$C_2$ via the photochemical oxidation of Pyr and $\omega C_2$ as intermediates (Lim et al., 2005),
whereas $C_3$ and $C_4$ diacids cannot be produced in this way.
In this study, we examined the seasonal variations of concentration ratios such as $C_2$/Pyr,
$C_2/\omega C_2$ and $C_2$/Gly to evaluate the oxidation strength of related precursors to the formation of
$C_2$. Relative high ratios of $C_2$/Pyr, $C_2/\omega C_2$ and $C_2$/Gly were observed in sampling seasons
except winter (Fig. 4c-e), but their values were much lower than those detected in particulate
matters at Gosan, Jeju Island (Kundu et al., 2010b), where the aerosols were relatively aged
during long-range transport. Figure S4 shows that strong relationships between $C_2/\omega C_2$,
$C_2$/Pyr and $C_2$/Tot (%) only existed in summer. No correlation was observed between $C_2$/Gly
and $C_2$/Tot (%). A negative correlation between $C_2/\omega C_2$ and $C_2$/Tot (%) in summer has never
been reported before. These phenomena demonstrate that abundant $C_2$, $\omega C_2$, Pyr, and Gly
were emitted directly from biogenic burning emissions, in the studied regions throughout the
year (Fig. 1). Only slightly stronger photochemical production of $C_2$ from Pyr was observed
in summer.
$C_3$ diacid can be produced as a result of hydrogen abstracted by OH radicals, followed by
decarboxylation processing of $C_4$ diacid (Kawamura and Ikushima, 1993). The mass
concentration ratio of $C_3/C_4$ is a good indicator for evaluating the contributions of
dicarboxylic acids from primary emissions or secondary oxidation production in the
atmosphere. Lower $C_3/C_4$ ratios were detected in vehicular exhaust by Kawamura and



362 Ikushima (1993), ranging from 0.25 to 0.44 with an average of 0.35. Less thermally stable $C_3$

363 diacid can degrade more preferentially to other species rather than remaining stable during

364 incomplete combustion processes.

365 Figure 4b shows that the $C_3/C_4$ ratios were relatively larger in the warm seasons (spring &

366 summer). However, the temporal trend of the $C_3/C_4$ ratios is relatively flat through the

367 sampling year. Most are less than or equal to unity, which is associated with the substantial

368 emissions from motor vehicles (Kawamura and Kaplan, 1987). The relatively low values of

369 the $C_3/C_4$ ratios caused by motor emissions were also detected in a previous study

370 (Kawamura and Kaplan, 1987). On the contrary, the prolonged secondary oxidation of

371 organic matters leads to $C_3/C_4$ values much greater than unity (Kawamura and Ikushima,

372 1993; Kawamura and Sakaguchi, 1999). The ratios of $C_3/C_4$ reported in this study are lower

373 than that (one-year average of 1.49) in urban Tokyo (Kawamura and Ikushima, 1993) and in

374 the remote Pacific Ocean (average 3) (Kawamura and Sakaguchi, 1999), where dicarboxylic

375 acids are largely produced by photooxidation reactions. These results demonstrated that in

376 addition to slightly enhanced atmospheric photochemical reactions in summer, incomplete

377 combustions overwhelmingly contributed to dicarboxylic acids in Beijing.

378 Phthalic acid (Ph) was one of the most abundant compounds during the sampling period. The

379 seasonal trends of phthalic acid to total dicarboxylic acids (Ph/Tot) are shown in Figure 4h.

380 The Ph/Tot ratios in winter were nearly 2–3 times greater than those in spring and autumn.

381 These findings imply that phthalic acid is largely emitted by anthropogenic sources in winter,

382 mainly as a result of intensive fossil fuel combustion. It is worth noting that Ph was abundant

383 in summer when increased ambient temperatures and stronger solar radiation facilitate the

384 transformation of gaseous PAHs (e.g., naphthalene) to produce relatively high levels of Ph.

385 $C_6$ and Ph are mostly formed via secondary oxidations of anthropogenic cyclic olefins (e.g.,

386 cyclohexene) and aromatic hydrocarbons, respectively, whereas $C_9$ is a photochemical

387 product of biogenic unsaturated fatty acids (Kawamura and Gagosian, 1987; Kawamura and

388 Ikushima, 1993). Thus, the mass concentration ratios of $C_6/C_9$ and Ph/$C_9$ may effectively

389 indicate the source strength of anthropogenic and biogenic emissions to these organic acids.





The seasonally averaged ratios of $C_6$/Tot, $C_9$/Tot, $C_6/C_9$ and Ph/$C_9$ are displayed in Table 3. Mean values of $C_6$/Tot are constantly low in all four seasons, whereas the seasonal ratios of $C_9$/Tot are the highest (0.09) in winter and the lowest (0.05) in summer, which result in the lowest value of $C_6/C_9$ ratios in winter (0.34±0.13), and are almost constant in the other three seasons. This trend is different from the one detected in the Central Himalayan region (1.07 in winter, 0.56 in summer) (Hegde and Kawamura, 2012) and Chennai, India (0.42 for winter, 0.29 for summer) (Pavuluri et al., 2010). In contrast, the values of Ph/$C_9$ are relatively high in winter (1.40±0.69) and summer (1.33±0.39), followed by spring (0.92±0.33) and autumn (0.82±0.39); its ratios are obviously lower than the values found in 14 other megacities in China (2.71 for winter, 3.37 for summer) (Ho et al., 2007) but are a bit higher than those in Tokyo (0.65 one year mean value) (Kawamura and Ikushima, 1993). From the outcomes discussed above, we concluded the contribution from anthropogenic emissions as the main source in megacities. Ph/$C_6$ ratios reached to the highest values in winter (4.06±0.78) and the lowest in autumn (1.66±0.78). A previous study demonstrated that the Ph/$C_6$ ratio from gasoline fuel vehicle (2.05) is lower than that from diesel fuel vehicles (6.58) (Kawamura and Kaplan, 1987). In addition, most of the Ph/$C_6$ values are larger than unity during the whole sampling year, which demonstrates that abundances of diacids attributable to more emissions from diesel burning than gasoline fuel vehicles.

Maleic acid (M), originated predominantly from photochemical oxidation of aromatic hydrocarbons (e.g., benzene and toluene), can be subsequently isomerized to its *trans*-isomer, fumaric acid (F), under favorable conditions. Lower M/F values have been detected in atmospheric aerosols over the North Pacific Ocean (0.3) (Kawamura and Sakaguchi, 1999) as well as at Alert in the high Arctic (ratio range: 0.5–1.0) (Kawamura et al., 1996a). The M/F ratios are almost constant in winter (2.0±0.66) and spring (2.0±0.67) and are higher than those in autumn (1.67±0.81) and summer (1.35±0.49). This trend may be associated with substantial amounts of precursors emitted from biomass burning in autumn, fresh aerosols brought by high-speed wind in spring and enhanced isomerization reaction from M to F under intense solar radiation in summer. The conversion of maleic to fumaric acids can be restrained in polluted environments with minimum weak sunlight (Kundu et al., 2010a). Thus,





M may not be effectively isomerized to F during wintertime in Beijing. The high ratios of
M/F throughout the whole year imply that aerosols in Beijing are not seriously subjected to
secondary oxidation processes.
Based on field observations, Kawamura and Ikushima (1993) hypothesized that $C_4$ diacid can
transform into malic acid ($hC_4$) by means of hydroxylation. The $hC_4/C_4$ ratios were the
highest in warm seasons (0.04±0.01 in summer, 0.03±0.02 in spring), which supported this
hypothesis. $hC_4/C_4$ ratios in summer are 2–4 times larger than those in cold seasons, similar
to the trends observed in Jeju Island, Korea (Kundu et al., 2010b) and urban Tokyo
(Kawamura and Ikushima, 1993).

**3.4  Comparisons of the mean mass ratios between sampling sites**

To assess the emission strength of anthropogenic activities in Beijing, the mean values of (a)
$C_3/C_4$, (b) M/F, (c) $Ph/C_9$, (d) Ph/Tot and (e) tPh/Tot mass ratios were compared with those in
other sampling sites, including Xi'an (Wang et al., 2012), Gosan, Jeju Island (Kundu et al.,
2010a) and the western Pacific Ocean (Wang et al., 2006b). Xi'an, a megacity in the
Guanzhong Plain, is located in one of the regions heavily polluted by fossil fuel and biofuel
combustion. Atmospheric aerosols at the Gosan site are mixtures of westerly winds from high
latitude regions of Eurasia. Marine aerosols over the western Pacific Ocean are a combination
of long-range transported continental aerosols and locally emitted marine aerosols.
Figure 5 presents the global distribution of diagnostic mass ratios of diacids and related
compounds. Rather low $C_3/C_4$ ratios were observed in urban aerosols, including Beijing and
Xi'an, compared to those aged organic matters collected from Gosan and the western Pacific
Ocean. Similarly, larger $C_3/C_4$ ratios were obtained in summer than in the other seasons. The
same observation in Beijing may be attributable to the enhancement of secondary oxidation
that favors the conversion of $C_4$ diacid to $C_3$ diacid in the warm season; however, in that case,
the photochemical activity is insignificant compared to the primary emissions. Similar to the
$C_3/C_4$ ratios, low M/F ratios indicate the importance of photochemical reaction routes
(Kawamura and Sakaguchi, 1999). The mean values of M/F in the Beijing aerosols are larger
than or comparable to those reported in Gosan (spring: 1.38, summer: 0.76, autumn: 1.62, and



winter: 2.21) but lower than those obtained in Xi'an aerosols (summer: 2.22 and winter: 2.38),
indicating that the PM$_{2.5}$ aerosols in Beijing are mainly linked with regional primary
emissions, whereas the photo-isomerization from *cis* to *trans* isomer is insignificant.
Usually, high Ph/C$_9$ ratios were detected in continental samples owing to a relatively strong
contribution from anthropogenic sources to dicarboxylic acids. A bit larger values of Ph/C$_9$
(in average) were obtained in Xi'an than those in Beijing because the air masses in Xi'an
were more heavily influenced by intense industrial emissions. Although the values of Ph/C$_9$
in both megacities were higher than those in the western Pacific Ocean, the wintertime Ph/C$_9$
ratios in Gosan were much greater than those in Beijing, which may be caused by the
secondary generation of abundant precursors, such as naphthalene, which were transported by
long-distance from East Asia.
In this study, we calculated the ratios (%) of Ph and tPh to total diacids, respectively, to
estimate the primary emission strength in different sampling sites. The largest mean mass
ratios of Ph/Tot were observed during winter in Beijing, while the values in the other seasons
were lower than those observed in Xi'an due to its basin-like topography. For the tPh/Tot
ratios, the mean values in Beijing were much higher than those in marine areas. However, the
average value of tPh/Tot in winter was lower than that in Xi'an. Thus, these comparisons
illustrate significant contributions from waste plastic burning and fossil fuel combustion in
Beijing during wintertime.
**3.5    Source identification by principal component analysis**
Previous studies have utilized principal component analysis (PCA) to discriminate the source
apportionment of atmospheric aerosols (Hopke, 1985). In this study, typical dicarboxylic
acids with other major components were chosen for factor analysis. Compounds with
common sources or photooxidation reactions would be likely to display similarities in mass
variations and be assorted into one "factor". High loadings of variables on the selected
species reveal closer links of sources and formation pathways between these compounds (Wu
et al., 2015). Here, "total varimax" maximizes the variance of the squared elements in the
columns of a factor matrix. The PCA result for dicarboxylic acids and other main



components in PM$_{2.5}$ in Beijing from Sep. 2013 to Jul. 2014 is given in Table 4.
During the whole sampling period, the first factor accounted for 75.2% of the total variance
with high loadings of selected diacids, WSOC, and EC (a tracer for incomplete
combustion-generated carbon emissions). Typically, the prolonged photochemical oxidation
of organics in the atmosphere leads to enhanced concentrations of polar organic matters.
WSOC can account for 45–75% of aerosol carbon mass in biomass burning emissions
(Falkovich et al., 2005) and 20–60% of that in fossil fuel combustion-derived particles
(Pathak et al., 2011). Agricultural waste burning is a substantial pollution factor in Beijing
(Fig. 1) (Viana et al., 2008; Cheng et al., 2014), especially in late June and early October,
resulting in substantial organic aerosols (Fu et al., 2012). C$_4$, C$_9$, tPh, ωC$_2$, Pyr, Gly and
MeGly showed strong correlations in the first factor, implying that burning activities
contribute to a large fraction of their concentrations, including biomass burning, biofuel
combustion and burning of municipal wastes. For example, the photooxidation of *p*-xylene, a
main precursor of terephthalic acid dimethyl ester, can produce glyoxal (Simoneit et al., 2005;
Kawamura and Pavuluri, 2010).
EC, maleic and phthalic acids are well associated with other species, indicating that they
originate from common mixed sources that are mainly produced by anthropogenic emissions,
such as vehicular exhaust, fossil fuel combustion and biomass burning. Aromatic
hydrocarbons from incomplete combustions are key materials for maleic and phthalic acids
(Kawamura and Sakaguchi, 1999). Both M and Ph showed abundances under hazy conditions
(Mochida et al., 2003).
As for the second factor, Ph, tPh and EC weakly loaded with each other, which seems to
originate from motor emissions, fossil fuel combustion and waste plastic burning. WSOC
also showed a slight loading in the second factor, which indicates that anthropogenic
emissions also contribute to a certain amount of WSOC during the sampling periods.

### 3.6    Stable carbon isotopic compositions

The systematic differences in stable carbon isotope ratios of diacids and other polar acids
were attributable to kinetic isotope fractionation processes in the atmosphere (Hoefs and



Hoefs, 1997), while secondary oxidation of these water-soluble organic acids is more
influential for diacid carbons to enrich in $^{13}$C (Wang and Kawamura, 2006c). For example,
the relatively short carbon-chain diacids enriched in $^{13}$C were ascribed to the kinetic isotopic
effect (KIE) for the photochemical breakdown of longer-chain diacids (Anderson et al., 2004;
Irei et al., 2006). And lower dicarboxylic acids with enrichment of $^{13}$C may be less active to
oxidants (e.g., OH radicals). Therefore, the determinations of $\delta^{13}$C values of dicarboxylic
acids and related compounds show vital information about the atmospheric aging processes
of aerosols derived from local emissions or long-range transport ways in air.
Table 5 presents the stable carbon isotope ratios of major compounds. The mean $\delta^{13}$C values
of $C_2$, $C_3$ and $C_9$ were constant among seasons, but those of $C_4$, $\omega C_2$, Pyr were smaller in
summer than in winter. Because coal is more enriched in $^{13}$C than that of petroleum fuel
(Court et al., 1981; Kawashima and Haneishi, 2012), the $^{13}$C enrichment of these organic
acids during wintertime may be attributable to the enhanced coal incineration for house
heating. Mean $\delta^{13}$C values of malonic acid in autumn and spring were similar to those of
succinic acid, suggesting that they may have similar sources or the same secondary formation
pathways.
The mean seasonal $\delta^{13}$C values of $C_9$ varying from −25.6‰ to −26.9‰ were smaller than
those of $C_2$–$C_4$ diacids. This signature demonstrates unsaturated fatty acids derived from
terrestrial vegetation as one key source of $C_9$, because more depletion of $^{13}$C in continental
higher plants in comparison with the particulate organic matters from marine plankton
activities. The $\delta^{13}$C values of $C_9$ suggested that azelaic acid is mainly from anthropogenic
primary emissions, especially biomass burning in the surrounding areas.
As mentioned earlier, Ph is mainly formed via the photochemical processes of polycyclic
aromatic hydrocarbons, but it can be emitted directly from fossil fuel combustion as well
(Kawamura and Kaplan, 1987; Fraser et al., 2003). The largest $\delta^{13}$C value of Ph in winter was
linked with its peak concentrations. This finding may be ascribed to the intensity of coal and
gasoline combustion in Beijing, especially the stagnant atmospheric conditions in favor of
accumulation of organic matters during wintertime (Cao et al., 2011). In general, the organic
aerosols derived from coal and gasoline burnings are more enriched in $^{13}$C than other



emissions, including diesel combustion, aerosols released from $C_3$-plants, and secondary
organic matters.
For terephthalic acid, the lowest $\delta^{13}C$ value of tPh (ave: –33.5‰) in winter supports the
finding that it is directly emitted from the burning of plastic wastes. Waste burning usually
contains many plastics and occurs frequently in open spaces without emission control
(Kawamura and Pavuluri, 2010), in addition to other local anthropogenic emissions. Lighter
$\delta^{13}C$ values of major compounds in Beijing than those in the marine and Arctic areas may be
explained by more contributions of primary emissions from anthropogenic sources.
Box plots of stable carbon isotope ratios ($\delta^{13}C$ values) are displayed in Fig. 6 for seasonal
distributions of diacids, glyoxylic and pyruvic acids in $PM_{2.5}$. There is a decreasing trend in
$\delta^{13}C$ for $C_5$ to $C_9$. Succinic acid showed the heaviest $\delta^{13}C$ value (–17.1‰) among all species
in winter and spring, while malonic acid was more enriched in $^{13}C$ than others in autumn (–
17.6‰) and summer (–18.7‰). Such trends were not observed for $C_3$ and $C_4$ diacids. A
previous study noted that increasing concentrations of oxalic and malonic acids inhibit the
growth of total fungi number due to the lower pH, which in turn changes the efficiency of
fungi to degrade the malonic acid (Côté et al., 2008). Hence, an enrichment of $^{13}C$ in
remaining malonic acid may be interpreted by the isotopic fractionations occurring in the
breakdown ways of dicarboxylic acids or photochemical degradation of $C_3$ diacid. In this
study, the median $\delta^{13}C$ values of $\omega C_2$ were much lower than those of $C_2$ in all seasons,
whereas $\delta^{13}C$ of Pyr showed median values similar to $C_2$ in autumn and spring, which are
surprisingly higher than that of $C_2$ in autumn.

### 3.7   Relations between $\delta^{13}C$ values and air mass source areas

In order to further estimate the impacts of air mass source regions on $\delta^{13}C$ values of specific
compounds, five-day backward trajectories for each aerosol sample are illustrated in Figure 1.
Data from urban Sapporo (Aggarwal and Kawamura, 2008), Gosan of the Jeju Island (Zhang
et al., 2016) and remote marine regions (Wang and Kawamura, 2006c) are plotted together
with the seasonal mean $\delta^{13}C$ values of major species detected in this study (Fig. 7). The
largest average $\delta^{13}C$ value of oxalic acid was observed in the Gosan samples. The seasonal



mean $\delta^{13}$C values of malonic acid in Beijing were higher than those in Sapporo and remote marine areas, but $C_3$ was less enriched in $^{13}$C compared to Gosan owing to the degradation of $C_3$ diacid or $C_2$ diacid depleted in $^{13}$C. The mean $\delta^{13}$C values of succinic acid are comparable to those in the other three places, except during summer. The mean $\delta^{13}$C values of Pyr in autumn (–19.6‰) and spring (–22.3‰) were similar to the data in Sapporo (–20.3‰) and Gosan (autumn: –19‰, winter: –22.2‰, spring: –19.1‰, summer: –17.6‰) aerosols. The $\delta^{13}$C values of $\omega C_2$ and Ph in remote marine samples are the highest, followed by those for Sapporo and Gosan sites and then Beijing. In contrast, the seasonal mean $\delta^{13}$C values of $C_6$ and $C_9$ in Beijing are similar to those in Sapporo and Gosan aerosols but lower than those in marine aerosols.

The air masses in Gosan, Jeju Island and Sapporo are mixtures of the flows from the mainland of East Asia. The $\delta^{13}$C values illustrate that organic aerosols in Sapporo are formed via photooxidation of precursors originated from anthropogenic and biogenic emissions (e.g., biomass burning) to a large extent, especially for $C_6$ and $C_9$; however, the study in Gosan found that aerosol samples are more aged in the western North Pacific rim. Most importantly, particulate organic matters in remote marine areas are intensively aged during long-range transport and are affected by both the sea-to-air emissions and the terrestrial outflows. Moreover, the enrichment in $^{13}$C can be regarded as a result of the isotopic fractionation for aged aerosols. Urban aerosols from Beijing, where the air masses are mixed with those originating from Siberia and surrounding areas, are seriously affected by biomass/biofuel burning in the whole year. Compared with the $\delta^{13}$C values in Gosan, Sapporo and remote marine areas, the smaller $\delta^{13}$C values of organic compounds in Beijing may be caused by the different emission strengths of various primary sources.

### 3.8 Relations between δ¹³C values and photochemical aging

$C_2$/Tot ratio is suggested to be a useful tracer to evaluate the aging of atmospheric aerosols (Kawamura and Sakaguchi, 1999). The mean $\delta^{13}$C values of oxalic acid showed the smallest value in winter (–22.9‰) and showed the highest value in autumn (–20.1‰), followed by spring (–21.9‰) (Fig. 8a). Here, we compared the $\delta^{13}$C values of $C_2$ and its concentration



changes with the relative abundance of $C_2$ to total diacids (Fig. 8b). The isotopic ratio values
of $C_2$ were positively correlated with $C_2$/Tot ratios in autumn ($r^2$=0.45) and winter ($r^2$=0.29),
suggesting that production of $C_2$ from the oxidation of precursors can contribute to the
increase of $\delta^{13}C$ values (Pavuluri et al., 2011). Due to enhanced primary emissions from coal
combustion and biomass burning, stagnant atmospheric inversion can favor the accumulation
of pollutants. Furthermore, the $\delta^{13}C$ values of tPh decreased from autumn to winter, followed
by an increase toward summer (Fig. 9). Seasonal $\delta^{13}C$ values of tPh decreased with the
enhanced ratios of tPh to total diacids (tPh/Tot) in autumn ($r^2 = 0.35$) and winter ($r^2 = 0.19$),
indicating large emissions from municipal waste burning activities in cold seasons, especially
in winter.
Aged organic aerosols are characterized by high abundance of polar and water-soluble
organic species, leading to high values of WSOC/OC ratio. However, in the Beijing samples,
the $\delta^{13}C$ values of major species ($C_2$, $C_3$, $C_4$, $C_9$, $\omega C_2$, Pyr, Ph and tPh) did not show strong
relationships with the WSOC/OC ratios in this paper (Fig. 10). The $\delta^{13}C$ values of $C_3$ only
correlated well with the WSOC/OC ratios in summer ($r^2 = 0.57$), in conformity with the
variation of $C_3$/$C_4$ ratios, which illustrates an enhanced degree of photochemical processing
of diacids during summertime. The $\delta^{13}C$ values of $C_4$ were negatively correlated with
WSOC/OC ratios in the cold seasons (autumn: 0.31, winter: 0.45), demonstrating an
enrichment of $\delta^{13}C$ in $C_4$ with decreasing WSOC/OC ratios. Ph displayed negatively weak
correlations in summer, while $\omega C_2$ presented weakly positive and negative relations in
autumn (0.2) and spring (0.29), respectively. The positive relationship between the $\delta^{13}C$
values of Pyr and WSOC/OC in autumn (0.62) suggests that an isotopic enrichment of Pyr
increases with high WSOC/OC ratios, which may have resulted in the largest $\delta^{13}C$ values of
Pyr in autumn. There are no correlations between $C_2$, tPh and $C_9$ with the WSOC/OC ratios in
the Beijing samples. Thus, the results discussed above suggest that primary emissions in local
regions significantly impact diacids and related compounds in Beijing.



## 4    Summary and conclusions

In this study, the molecular distribution and stable carbon isotopic composition of diacids, oxoacids, and α-dicarbonyls were determined in fine aerosol samples (PM$_{2.5}$) in Beijing over one year. Oxalic acid was found to be the most abundant diacid throughout the year. The concentration patterns of major identified organic compounds varied among different seasons. Such differences in molecular compositions were caused by diverse emission strengths of primary emission sources together with photooxidation processes in Beijing. Correlation analyses of main oxoacids and α-dicarbonyls with combustion tracers (Cl$^-$ and K$^+$) indicate that ωC$_2$, Pyr, Gly and MeGly were mostly affected by biogenic combustions in whole sampling year, with significant contribution of fossil fuel combustion in winter. The variations in the C$_3$/C$_4$ ratios were relatively minor during the one-year observation, with most values less than or equal to unity, which is associated with the substantial emissions from vehicular exhausts. Higher ratios of Ph/Tot and tPh/Tot were observed in winter, indicating strong influences of fossil fuel combustion and burning of plastic waste.

Larger δ$^{13}$ values obtained in lower carbon-numbered diacids are mainly interpreted as isotopic fractionations due to the decomposition of longer-chain dicarboxylic acids and related precursors. Although oxalic acid has been regarded as a final product of the photooxidation of homologues diacids and related components like Pyr, ωC$_2$ and α-dicarbonyls in the atmosphere, succinic acid showed the largest δ$^{13}$C value (–17.1‰) among all the species in winter and spring, while malonic acid was more enriched in $^{13}$C than others in autumn (–17.6‰) and summer (–18.7‰). The less negative δ$^{13}$C value of malonic acid may be interpreted by the isotopic fractionations occurring in the breakdown of diacids or photochemical degradation of C$_3$ diacid.

On the basis of the weak correlations of C$_2$/Tot and WSOC/OC with seasonal δ$^{13}$C values of major species, the results of the principal component analysis, and the comparison of δ$^{13}$C values in Beijing with those in urban and remote marine aerosols, we can conclude that photochemical production of dicarboxylic acids and related compounds in the Beijing aerosols was insignificant in the whole sampling period. The abundance of diacids and




related polar acids in fine aerosols in Beijing are mainly associated with anthropogenic primary emissions such as biomass burning, fossil fuel combustion and plastic burning. Further study is needed to interpret the detailed mechanisms of the enrichment of the $\delta^{13}C$ values of $C_3$ and $C_4$ diacids and to better evaluate the impact of micro-biological degradation along with contact-induced chemical changes on the aerosol chemistry in Beijing.

## Acknowledgements

This study was supported by the National Natural Science Foundation of China (Grant Nos. 41475117, 41571130024 and 91543205) and the Strategic Priority Research Program (B) of the Chinese Academy of Sciences (Grant No. XDB05030306). P.F. thanks the financial support from the National Science Fund for Distinguished Young Scholars (Grant No. 41625014).



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





**Table 1.** Seasonal concentrations (ng m$^{-3}$) of dicarboxylic acids, ketocarboxylic acids and
α-dicarbonyls in PM$_{2.5}$ samples collected in Beijing from 30 September 2013 to 12 July 2014.

| Species (Abbr.) | Autumn (n=16) | | Winter (n=15) | | Spring (n=19) | | Summer (n=15) | |
| --- | --- | --- | --- | --- | --- | --- | --- | --- |
| | Range | Mean/SD | Range | Mean/SD | Range | Mean/SD | Range | Mean/SD |
| **Dicarboxylic acids** | | | | | | | | |
| Oxalic, C$_2$ | 31–1760 | 472/490 | 44.9–456 | 149/123 | 96.5–496 | 262/120 | 64.7–462 | 267/146 |
| Malonic, C$_3$ | 6.0–132 | 43.5/36.1 | 5.8–54.2 | 20.1/15.6 | 8.4–64.9 | 33.0/14.2 | 13.9–46.9 | 30.5/13.0 |
| Succinic, C$_4$ | 11.5–231 | 67.2/62.1 | 11.1–81.0 | 31.6/21.3 | 11.4–82.0 | 37.7/17.4 | 14.5–54.8 | 31.2/14.0 |
| Glutaric, C$_5$ | 2.8–50.3 | 15.2/13.6 | 3.5–20.9 | 9.2/5.6 | 4.9–17.8 | 10.3/3.8 | 4.4–13.9 | 8.8/3.4 |
| Adipic, C$_6$ | 4.4–38.8 | 16.2/9.0 | 2.9–19.0 | 8.9/5.0 | 5.9–21.1 | 13.6/3.8 | 4.9–16.7 | 10.6/4.0 |
| Pimeric, C$_7$ | 0.8–16.7 | 6.0/6.4 | 0.6–11.4 | 3.4/3.4 | 1.7–7.4 | 3.9/1.8 | 1.1–5.2 | 3.0/1.2 |
| Suberic, C$_8$ | BDL–24.3 | 4.7/7.3 | BDL | BDL | BDL–10 | 2.3/3.3 | BDL–5.1 | 0.8/1.6 |
| Azelaic, C$_9$ | 13.7–59.3 | 31.6/14.2 | 12.1–60.3 | 27.3/14.7 | 15.1–60 | 27.2/11.1 | 11.0–28.2 | 19.0/5.0 |
| Decanedioic, C$_{10}$ | 0.2–7.7 | 2.3/2.0 | 0.4–2.6 | 1.2/0.6 | 0.7–3.2 | 1.6/0.8 | 0.9–3.0 | 1.7/0.6 |
| Undecanedioic, C$_{11}$ | 0.4–10.0 | 2.7/2.4 | 0.6–5.7 | 2.2/1.6 | 1.1–3.1 | 2.0/0.6 | 1.1–2.4 | 1.8/0.5 |
| Dodecanedioc, C$_{12}$ | BDL–2.1 | 0.5/0.5 | BDL–1.8 | 0.1/0.5 | BDL–0.5 | 0.2/0.2 | BDL–0.5 | 0.2/0.2 |
| Methylmalonic, iC$_4$ | 0.1–3.3 | 1.1/0.8 | 0.3–2.3 | 1.0/0.6 | 0.5–3.0 | 1.1/0.6 | 0.5–1.9 | 0.9/0.4 |
| Methylsuccinic, iC$_5$ | 1.3–24.7 | 7.3/6.8 | 2.2–14.5 | 5.7/3.8 | 1.4–6.8 | 3.8/1.9 | 0.7–3.8 | 2.2/0.9 |
| 2-methylglutaric, iC$_6$ | 0.2–6.6 | 1.8/1.8 | 0.3–2.9 | 1.1/0.7 | 0.4–1.8 | 1.0/0.5 | 0.3–1.3 | 0.8/0.4 |
| Maleic, M | 1.0–12.6 | 3.7/3.1 | 1.2–6.6 | 3.0/1.6 | 1.1–6.3 | 2.5/1.4 | 1.0–3.3 | 1.8/0.7 |
| Fumaric, F | 0.4–11.3 | 3.0/3.0 | 0.4–4.5 | 1.8/1.5 | 0.5–3.0 | 1.4/0.8 | 0.7–2.6 | 1.5/0.7 |
| Methylmaleic, mM | 1.1–17.3 | 5.2/4.7 | 1.7–11.7 | 4.8/3.1 | 1.3–5.8 | 2.5/1.6 | 0.8–4.6 | 2.2/1.1 |
| Phthalic, Ph | 7.6–58.7 | 25.5/15.8 | 11.4–98.5 | 37.9/27.2 | 8.5–36.7 | 22.5/7.1 | 13.4–42.3 | 24.9/8.0 |
| Isophthalic, iPh | 0.5–6.2 | 1.9/1.6 | 0.5–4.2 | 1.8/1.2 | BDL–2.6 | 0.7/0.6 | 0.3–1.1 | 0.8/0.3 |
| Terephthalic, tPh | 8.9–80.4 | 40.3/25.0 | 10.8–136 | 48.7/41.1 | 4.6–35.3 | 19.5/9.3 | 5.2-26.0 | 15.5/6.0 |
| Malic, hC$_4$ | BDL–6.5 | 1.3/2.0 | BDL–0.8 | 0.2/0.3 | 0.4–4.5 | 1.2/1.3 | 0.5-4.0 | 1.2/1.0 |
| Oxomalonic, kC$_3$ | 0.7–24.2 | 6.8/6.7 | 1.3–18.0 | 5.0/4.7 | 0.8–12.7 | 6.5/3.5 | 1.1–8.7 | 4.2/2.4 |
| 4-oxopimelic, kC$_7$ | 0.3–8.8 | 3.0/2.5 | 0.3–5.8 | 1.6/2.1 | 0.8–7.2 | 3.2/1.7 | 1.3–10.2 | 4.7/2.9 |
| Total diacids | 110–2580 | 763/701 | 113–1010 | 366/261 | 158–781 | 460/180 | 171–722 | 435/195 |
| **Oxocarboxylic acids** | | | | | | | | |
| Pyruvic, Pyr | 2.0–56.0 | 15.6/14.9 | 2.6–68.7 | 13.5/17.6 | 4.5–21.7 | 11.5/5.3 | 3.6–19.3 | 10.9/6.0 |
| Glyoxylic, ωC$_2$ | 3.3–183 | 43.7/50.4 | 6.9–275 | 44.3/69.0 | 7.3–61.1 | 25.1/15.3 | 4.0–49.7 | 24.7/17.0 |
| 3-oxopropanoic, ωC$_3$ | 0.6–23.5 | 6.0/6.2 | 0.8–23.1 | 5.6/6.2 | 1.0–8.2 | 4.7/2.2 | 1.4–7.2 | 3.7/1.7 |
| 4-oxobutanoic, ωC$_4$ | 2.1–41.3 | 11.9/10.6 | 2.9–32.2 | 10.5/9.0 | 3.0–14.2 | 8.0/3.5 | 1.9–12.1 | 6.5/3.3 |
| 5-oxopentanoic, ωC$_5$ | 0.7–8.2 | 2.7/2.1 | 0.8–6.7 | 2.5/1.7 | 0.8–4.1 | 2.2/0.9 | 0.7–3.5 | 1.8/0.9 |
| 7-oxoheptanoic, ωC$_7$ | 0.5–7.0 | 3.0/2.0 | 0.4–5.0 | 1.9/1.6 | 1.0–4.8 | 3.1/1.0 | 1.6–6.9 | 3.5/1.5 |
| 8-oxooctanoic, ωC$_8$ | 0.4–12.3 | 4.0/3.3 | 0.2–9.2 | 2.4/2.7 | 0.4–6.6 | 3.0/1.4 | 2.4–9.2 | 5.2/2.4 |
| 9-oxononanoic, ωC$_9$ | 0.4–7.2 | 2.0/1.8 | 0.6–2.9 | 1.6/0.8 | 0.3–1.9 | 1.1/0.4 | 0.2–2.2 | 1.1/0.6 |
| Total ketoacids | 9.5–282 | 73.3/76.3 | 13.5–353 | 68.7/91.0 | 14.5–95.0 | 47.3/24.6 | 15.1–82.8 | 46.5/27.0 |
| **α-dicarbonyls** | | | | | | | | |
| Glyoxal, Gly | 0.6–36.6 | 9.3/10.8 | 1.5–31.0 | 7.2/8.1 | 1.8–9.8 | 4.2/2.3 | 0.9–7.9 | 3.8/2.5 |
| Methylglyoxal, MeGly | 1.0–49.3 | 15.9/17.3 | 1.5–30.9 | 8.3/7.9 | 1.5–26.1 | 8.5/6.8 | 1.7–22.3 | 9.0/7.2 |
| Total dicarbonyls | 1.5–85.9 | 25.1/28.1 | 3.7–61.9 | 15.5/15.9 | 3.9–35.9 | 12.7/9.1 | 2.6–30.1 | 12.7/10.0 |

BDL: below detection limit, which is ca. 0.005 ng m$^{-3}$ for the target compounds.


Atmospheric Chemistry and Physics Discussions — Open Access — EGU

**Table 2.** Comparison of characteristics of diacids at the Beijing site and its other areas detected from previous studies.

| Location | Sampling Date | Size | Diacid (C₂-C₁₂) Concentrations (Mean) ng m⁻³ | Major Species | Diacid-C/OC (Diacid-C/TC) % | WSOC/OC (WSOC/TC) % | Reference |
|---|---|---|---|---|---|---|---|
| Fourteen Chinese cities | Jun-Jul 2003 | $PM_{2.5}$ | 211–2162 (892) | $C_2$>Ph>$C_4$>$C_3$ | 2.3 (1.4[b]) | 48 (37) | Ho et al. [2007] |
| Fourteen Chinese cities | Jan 2003 | $PM_{2.5}$ | 319–1940 (904) | $C_2$>$C_4$>Ph>$C_3$ | 1.3 (1.0[b]) | 41 (32) | Ho et al. [2007] |
| Xi'an, China | Jan-Feb 2009 | $PM_{10}$ | 1033–2653 (1843) | $C_2$>tPh>Ph>o$C_2$ | 1.1[b] (0.83) | 54[b] (41[b]) | Cheng et al. [2013] |
| Xi'an, China | Aug 2009 | $PM_{10}$ | 478–2040 (1259) | $C_2$>Ph>$C_4$>$C_3$ | 4.4[b] (3.8[b]) | 80[b] (52[b]) | Cheng et al. [2013] |
| Beijing, China | Sep-Oct 2007 | TSP | 105–3056 (1208) | $C_2$>$C_4$>$C_3$>Ph | (3.0) | | He et al. [2014] |
| Hong Kong, China | Aug 2003 | $PM_{2.5}$ | 260–677 (454) | $C_2$>Ph>iPh>tPh | 0.41 | | Wang et al. [2006a] |
| Hong Kong, China | Feb 2004 | $PM_{2.5}$ | 114–812 (771) | $C_2$>Ph>iPh>tPh | 0.51 | | Wang et al. [2006a] |
| Sapporo, Japan | May-Jul 2005 | TSP | 106–787 (406) | $C_2$>$C_3$>$C_4$>Ph | 4.8 (1.8) | 44 (39) | Aggarwal and Kawamura [2008] |
| Chennai, India | Jan-Feb, May 2007 | $PM_{10}$ | 176–1436 (612) | $C_2$>$C_3$>$C_4$>tPh | (1.6) | | Pavuluri et al. [2010] |
| Gosan, South Korea | Apr 2003-Apr 2004 | TSP | 142–1875 (636) | $C_2$>$C_3$>$C_4$>o$C_2$ | | | Kundu et al. [2010b] |
| Tokyo, Japan | Apr 1988-Feb 1989 | TSP | 90–1360 (480) | $C_2$>$C_3$>$C_4$>$C_9$ | (0.95) | | Kawamura and Ikushima [1993] |
| Ulaanbaatar, Mongolia | Nov 2007-Jan 2008 | $PM_{2.5}$ | 146–779 (536) | tPh>$C_2$>$C_4$>Ph | 0.8 (0.6) | 53.2 (43.8) | Jung et al. [2010] |
| Chengdu, China | Jan 2013 (Daytime) | $PM_{2.5}$ | 1490–4690 (3450) | $C_2$>$C_4$>Ph>tPh | 2.3[b] (4.8[b]) | | Li et al. [2015] |
| Chengdu, China | Jan 2013 (Nighttime) | $PM_{2.5}$ | 1410–5250 (3330) | $C_2$>$C_4$>Ph>tPh | 2.2[b] (4.2[b]) | | Li et al. [2015] |
| Beijing, China | Sep 2013-Jul 2014 | $PM_{2.5}$ | 110–2580 (506) | $C_2$>$C_4$>o$C_2$>$C_3$ | 2.0[b] (1.5[b]) | 60[b] (45[b]) | This study |

[b]Calculated from the mean values from the references.



**Table 3.** Average seasonal variations in the ratios of diacids and related compounds.

| Ratios | Autumn | Winter | Spring | Summer |
|---|---|---|---|---|
| $C_2$/Total diacids (%) | 0.54±0.12 | 0.39±0.05 | 0.56±0.07 | 0.58±0.1 |
| $C_3$/$C_4$ ratio | 0.69±0.14 | 0.59±0.11 | 0.88±0.09 | 0.99±0.1 |
| $C_6$/$C_9$ ratio | 0.53±0.24 | 0.34±0.13 | 0.55±0.21 | 0.59±0.28 |
| Ph/$C_9$ ratio | 0.82±0.39 | 1.4±0.69 | 0.92±0.33 | 1.33±0.39 |
| Ph/$C_6$ ratio | 1.7±0.78 | 4.1±0.78 | 1.8±0.85 | 2.5±0.76 |
| M/F ratio | 1.7±0.81 | 2.0±0.66 | 2.0±0.67 | 1.4±0.49 |
| $hC_4$/$C_4$ ratio | 0.01±0.01 | 0.01±0.01 | 0.03±0.02 | 0.04±0.01 |
| $C_9$/Total diacids (%) | 0.07±0.06 | 0.09±0.03 | 0.07±0.03 | 0.05±0.02 |
| Ph/Total diacids (%) | 0.04±0.02 | 0.11±0.01 | 0.05±0.02 | 0.07±0.03 |
| $C_6$/Total diacids (%) | 0.03±0.02 | 0.03 | 0.03±0.01 | 0.03±0.02 |
| WSOC/OC ratio | 0.70±0.27 | 0.49±0.11 | 0.56±0.07 | 0.58±0.10 |




**Table 4.** Results of the principal component analyses for selected diacids and related
compounds in PM$_{2.5}$ in Beijing.

| Species | Whole year | |
|---|---|---|
| | Factor 1 | Factor 2 |
| C$_2$ | 0.87 | |
| C$_3$ | 0.89 | |
| C$_4$ | 0.92 | |
| C$_6$ | 0.74 | |
| C$_9$ | 0.72 | |
| M | 0.94 | |
| F | 0.96 | |
| Ph | 0.76 | 0.51 |
| tPh | 0.78 | 0.55 |
| ωC$_2$ | 0.92 | |
| Pyr | 0.92 | |
| Gly | 0.97 | |
| MeGly | 0.88 | |
| WSOC | 0.89 | 0.38 |
| EC | 0.81 | 0.46 |
| Total variance | 75.2% | 10.9% |


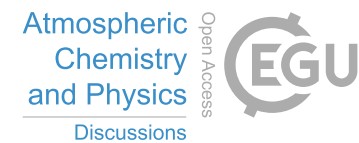

**Table 5.** Stable carbon isotope ratios (δ13C, ‰) of major compounds in PM2.5 in Beijing.

| Species | Autumn | | | Winter | | | Spring | | | Summer | | |
|---|---|---|---|---|---|---|---|---|---|---|---|---|
| | Min | Max | Mean±SD | Min | Max | Mean±SD | Min | Max | Mean±SD | Min | Max | Mean±SD |
| C2 | −23.7 | −15 | −20.1±3.0 | −27.2 | −14.8 | −22.9±3.4 | −25 | −16.6 | −21.9±2.1 | −27 | −19.1 | −22.4±2.7 |
| C3 | −27.2 | −12.3 | −17.6±4.6 | | | | −25.2 | −5.6 | −17.3±8.6 | −24 | −12.6 | −18.7±4 |
| C4 | −25.2 | −15.8 | −19.8±2.3 | −22.1 | −9.8 | −17.1±3.9 | −20.6 | −13.1 | −17.1±2.0 | −37.6 | −19.4 | −28.6±6.8 |
| C5 | −22.9 | −31.8 | −25.9±2.6 | −29.7 | −26.3 | −28±1.5 | −28.4 | −23.5 | −25.4±1.9 | | | |
| C6 | −43.2 | −22.4 | −28.5±8.5 | | | | −28.1 | −24.5 | −26.1±1.6 | | | |
| C9 | −31.3 | −21.4 | −26.4±2.6 | −28.2 | −23.7 | −25.6±1.4 | −31.3 | −23.1 | −26.9±2.0 | −28.6 | −20.8 | −25.9±2.4 |
| ωC2 | −47.1 | −26.3 | −37.2±7.3 | −61.6 | −19.6 | −32±13.7 | −44 | −19.4 | −32.5±7.6 | −44.3 | −18.8 | −32.2±7.3 |
| Pyr | −29.1 | −14.8 | −19.6±5.9 | −32.8 | −16.7 | −27.3±9.2 | −38.1 | −12.6 | −22.3±9.7 | −62.4 | −19.0 | −38.1±16 |
| Ph | −47.5 | −25.3 | −32.6±7.9 | −30.7 | −25.1 | −27.7±1.5 | −33 | −25.1 | −28.3±1.9 | −32.5 | −27.6 | −30.4±1.3 |
| tPh | −27.6 | −23.6 | −25.2±1.0 | −40.1 | −28.5 | −33.5±3.4 | −33.4 | −22.8 | −25.8±3 | −26.1 | −18.9 | −23.5±2.3 |






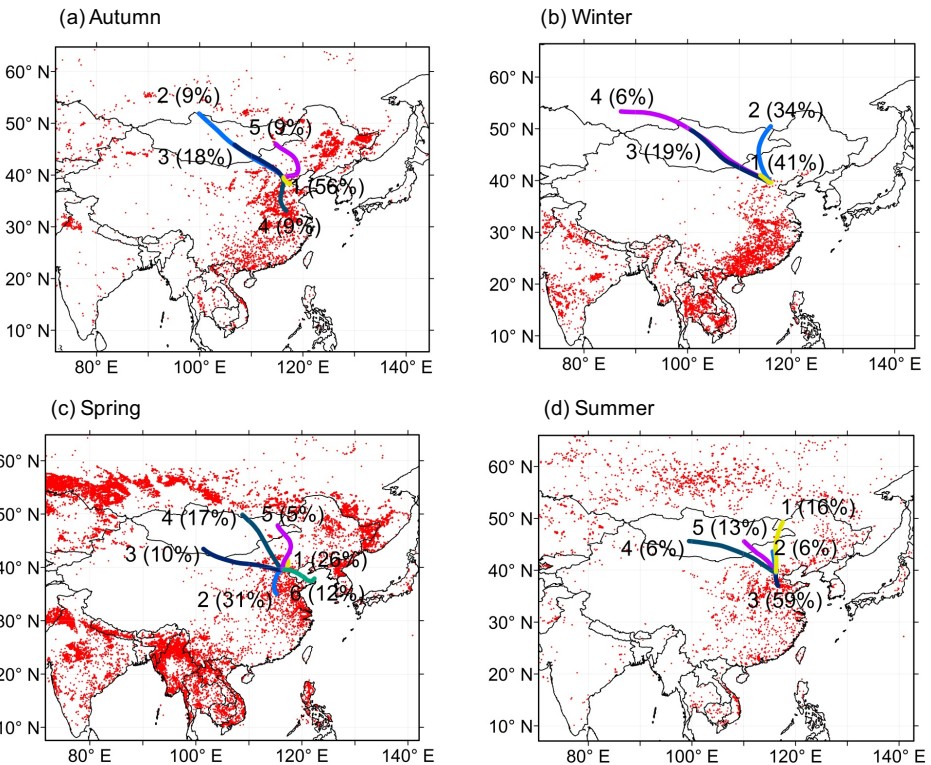

**Figure 1.** Fire spots with typical five-day air mass backward trajectories (mean clusters) arriving at Beijing for each sampling season. The fire spot data were obtained from the MODIS fire spot website (https://firms.modaps.eosdis.nasa.gov/download/request.php). The air mass trajectories were drawn using the data obtained by HYSPLIT4 model from the NOAA ARL website (http://ready.arl.noaa.gov/HYSPLIT.php). The arrival height of the air mass backward trajectories was 500 m above sea level.





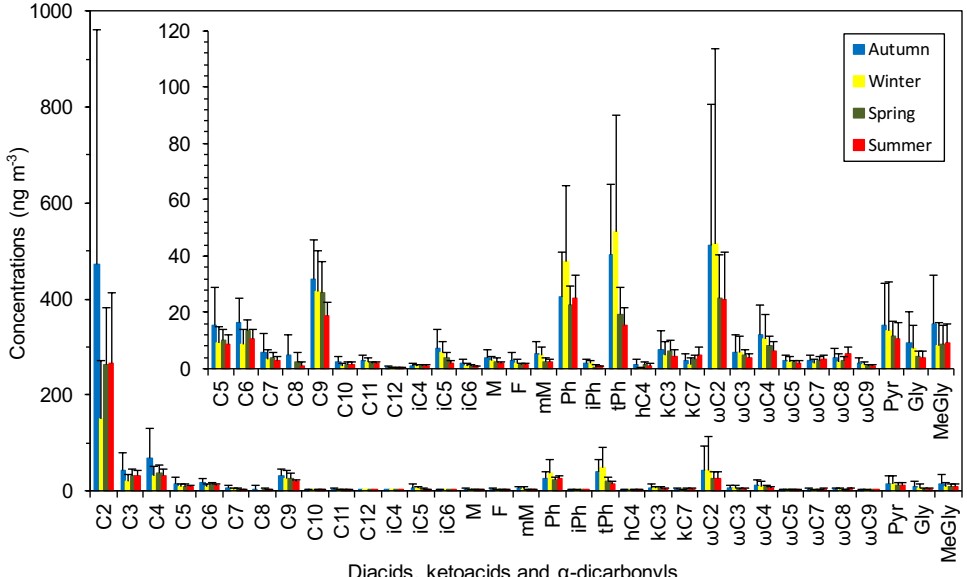



**Figure 2.** Molecular distributions of dicarboxylic acids and related compounds in the PM$_{2.5}$

samples collected in Beijing from 30 September 2013 to 12 July 2014.




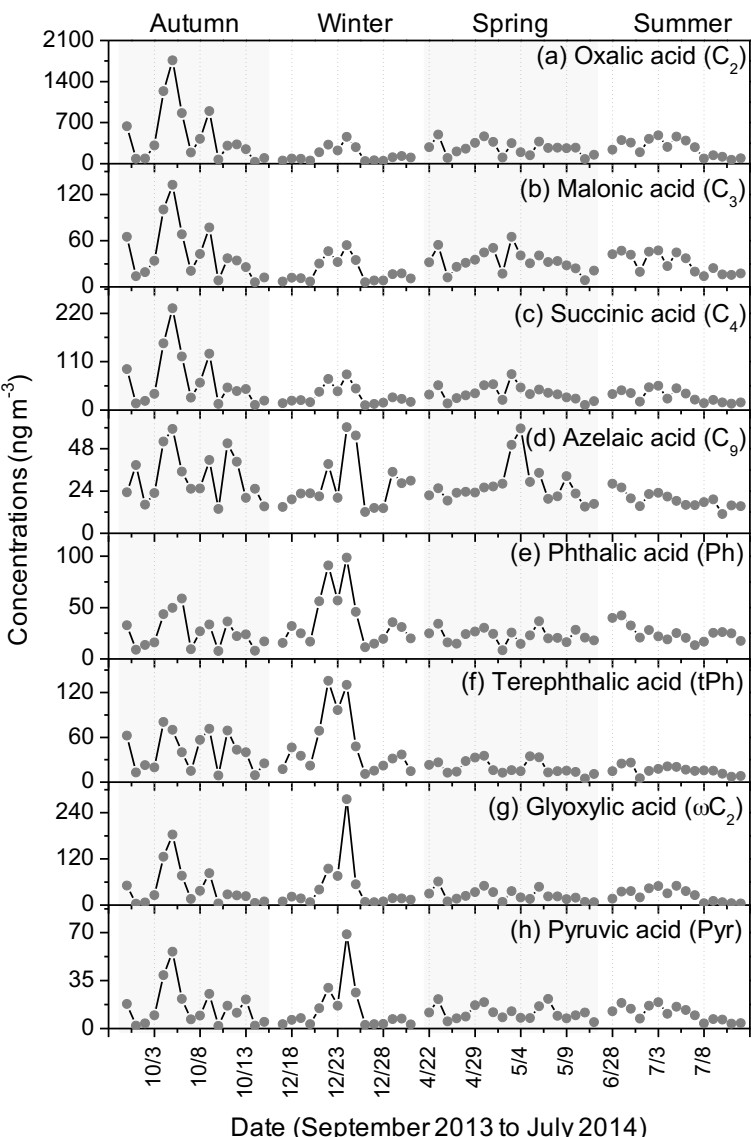



**Figure 3.** Daily variations in the concentrations of selected organic acids in the $PM_{2.5}$ aerosols
in Beijing.



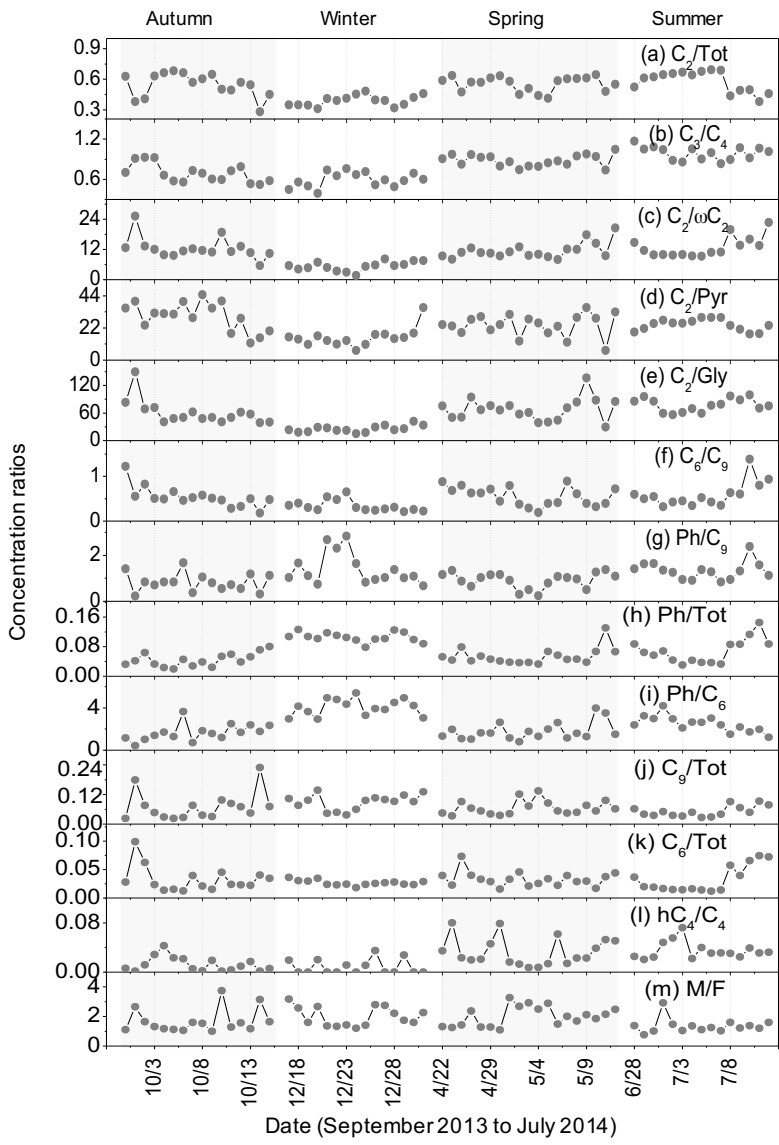



**Figure 4.** Seasonal variations in the concentration ratios of (a) $C_2$/Tot, (b) $C_3$/$C_4$, (C) $C_2$/$\omega C_2$, (d) $C_2$/Pyr, (e) $C_2$/Gly, (f) $C_6$/$C_9$, (g) Ph/$C_9$, (h) Ph/Tot, (i) Ph/$C_6$, (j) $C_9$/Tot, (k) $C_6$/Tot, (l) $hC_4$/$C_4$, and (m) M/F in the Beijing aerosols.






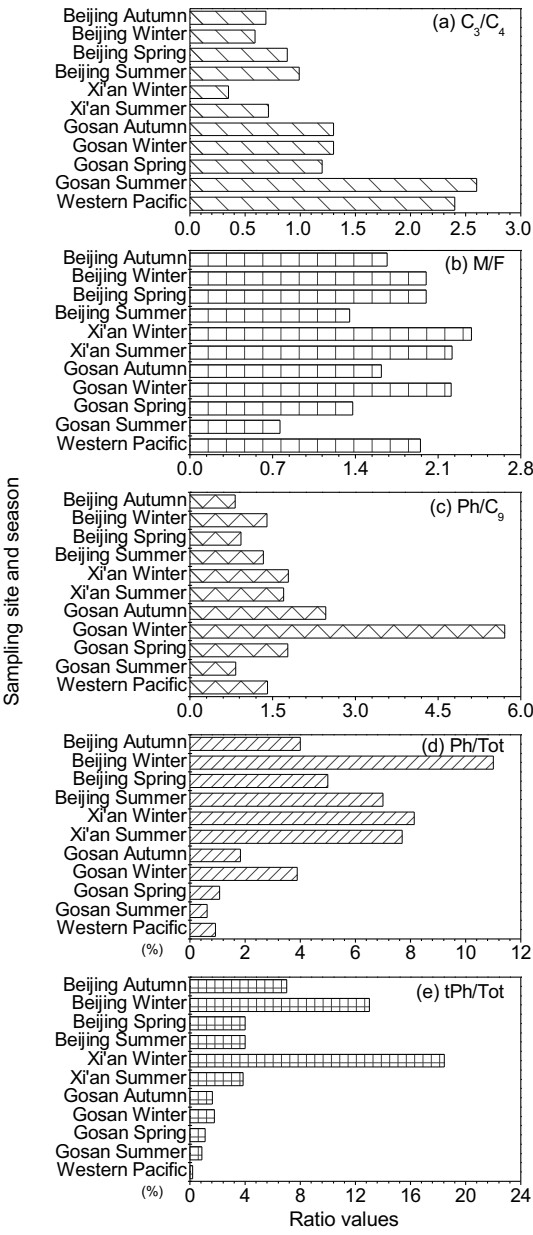

**Figure 5.** Mean mass ratios of (a) $C_3/C_4$, (b) M/F, (c) $Ph/C_9$, (d) Ph/Tot, and (e) tPh/Tot from this study compared with those in Xi'an (Wang et al., 2012), Gosan, Jeju Island (Kundu et al., 2010a) and the western Pacific (Wang et al., 2006b) aerosols.





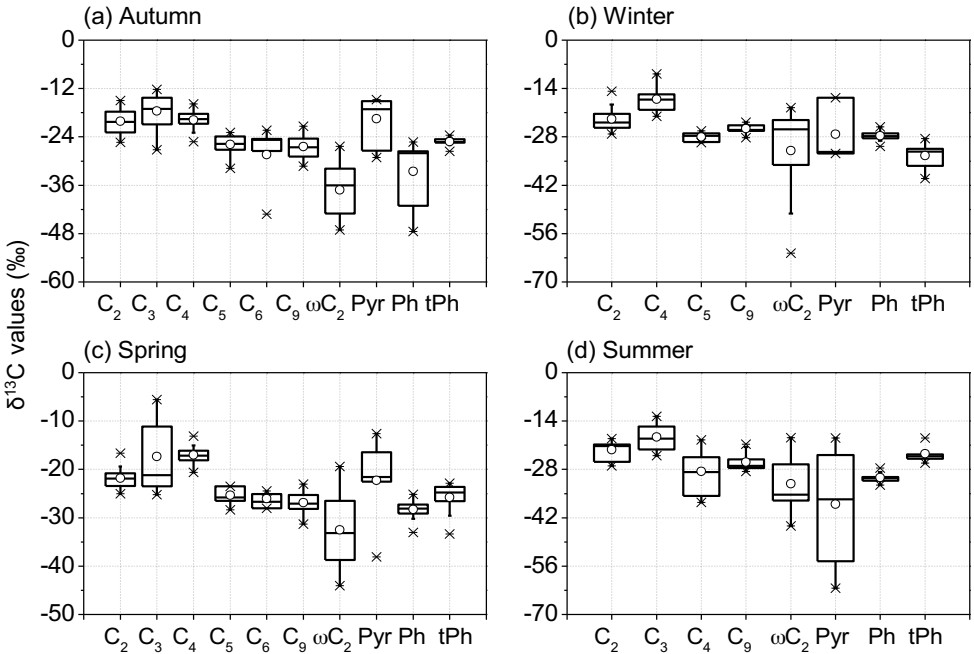



**Figure 6.** Box plot of the $\delta^{13}C$ values of diacids, glyoxylic and pyruvic acids. The small circles
represent the average $\delta^{13}C$ values.



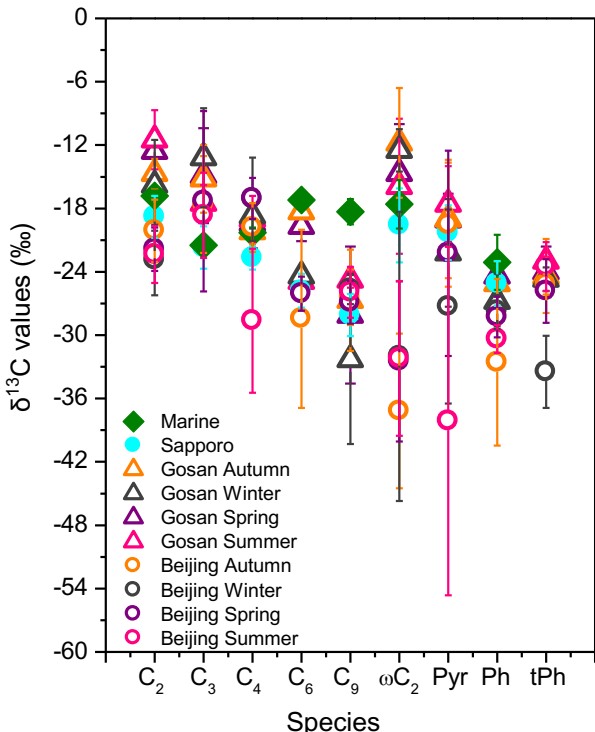



**Figure 7.** Seasonal mean $\delta^{13}C$ values of selected diacids and related compounds detected in
PM$_{2.5}$ in Beijing. Data from Saporro (Aggarwal and Kawamura, 2008), Gosan, Jeju Island
(Zhang et al., 2016) and marine (Wang and Kawamura, 2006c) aerosols are also plotted. The
bar represents the standard variation (±SD) in the $\delta^{13}C$ values.





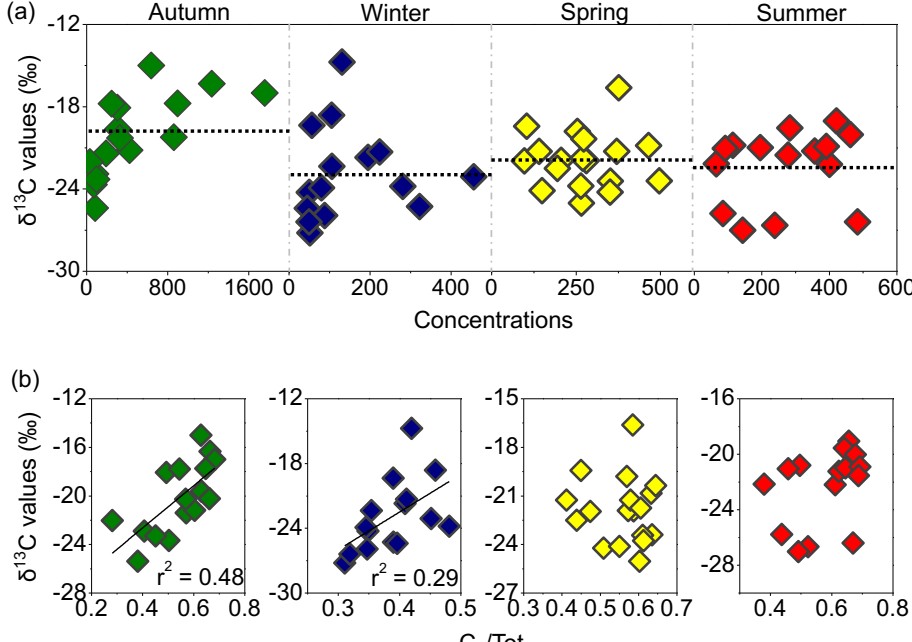



**Figure 8.** (a) Seasonal variations in the stable carbon isotope ratios ($\delta^{13}C$) of $C_2$, (b) correlations between $\delta^{13}C$ values of $C_2$ and relative abundances of oxalic acid to total diacids ($C_2$/Tot) in $PM_{2.5}$ in Beijing. The black dotted lines represent the average $\delta^{13}C$ values.










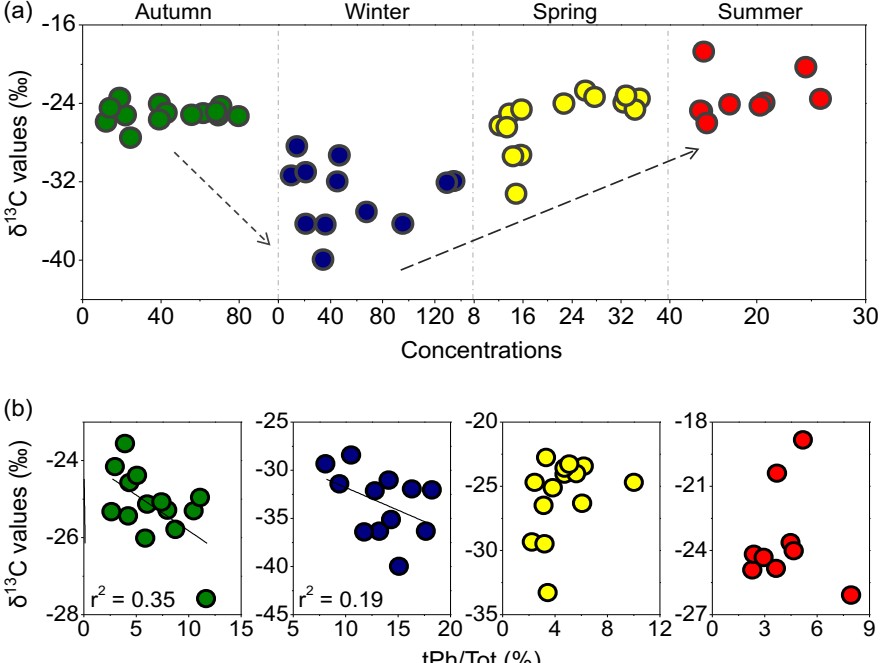



**Figure 9.** (a) Seasonal variations in stable carbon isotope ratios ($\delta^{13}C$) of tPh, (b) correlations
between the $\delta^{13}C$ values of tPh and relative abundances of terephthalic acid to total diacids
(tPh/Tot) in $PM_{2.5}$ in Beijing.








**Figure 10.** Correlations between compound-specific stable carbon isotope ratios of selected

diacids and oxoacids and WSOC/OC ratios in PM$_{2.5}$ in Beijing.