# Peer review of "Molecular distribution and compound-specific stable"

_Atmospheric Chemistry and Physics, 2017_

## Referee Comment (RC1) · Anonymous Referee #2 · 16 Aug 2017

Review of "Molecular distribution and compound-specific stable carbon isotopic composition of dicarboxylic acids, oxocarboxylic acids, and α-dicarbonyls in $PM_{2.5}$ from Beijing, China" by Zhao et al.

In this study, the authors measured the concentrations and stable carbon isotopic composition of dicarboxylic acids, oxocarboxylic acids, and α-dicarbonyls in Beijing, China. The annual variation of species, mass ratios between species, and $\delta^{13}C$ are used to infer the sources and formation processes of measured species. The authors found that (1) anthropogenic primary emissions are the major contribution to the measured species and (2) the photooxidation is weak and contributes insignificant amount of measured species. However, I find the conclusions are not well justified. Particularly, many discussions suffer flaws in its logical flow and it is unclear how the conclusions are drawn from the analysis/evidence. also, some discussions contradict with each other. Therefore, I would not recommend this manuscript for publication in its current state.

Major Comments

1.      Many discussions contradict with each other. This causes severe confusion and leaves the conclusions ambiguous. One major conclusion in this study is that "the degree of photochemical formation of diacids in Beijing is insiginificant". However, there are many discussions in the manuscript which contradict with this conclusion. For example, Line 337-339. The authors state that "C2/Tot showed strong correlations with C2/C4 in all four seasons, indicating the **significance of photooxidation pathways** of biogenic unsaturated fatty acids".  This sentence directly contradicts the main conclusion of this study.

In Line 419-421, it is stated that "the high ratios of M/F throughout the whole year imply that aerosols in Beijing are not seriously subjected to secondary oxidation process". While the photooxidation may be weak in winter, it is strong in summer. The statement in Line 419-421 directly contradicts to many studies (for example, those cited in the introduction section). How do authors reconcile the discrepancies?

Some other contradictions are listed below. This is by no means an exclusive list. The authors must check carefully about inconsistency throughout the manuscript.

In Line 381, "these findings imply that Ph is largely **emitted** by anthropogenic sources in winter". In line 385, "Ph are mostly formed via **secondary oxidations** of anthropogenic aromatic hydrocarbons."

In Line 239-242 and Line 257, "…indicating that substantial amounts of C9 may be stemmed from the **local and surrounding combustion activities** in Beijing." Line 386, "C9 is a photochemical product of **biogenic** unsaturated fatty acids." Then, C6/C9 is used to evaluate the source strength of anthropogenic vs biogenic emissions.

2.     Many discussions suffer flaws in logical flow and the link between evidence and conclusion is not clear. To name a few,

(1) Line 43-46. Why could "lower $\delta^{13}C$ of major species in Beijing than western North Pacific" indicate "weak photooxidation in Beijng"? What is the rationale to compare Beijing with western North Pacific?

(2) Line 258-268. Firstly, the authors mentioned that phthalic acids (Ph) can be formed vis photooxidation of naphthalene or directly emitted by fossil fuel burning and incomplete combustion. Secondly, the authors stated that the great amount of naphthalene is the precursor of Ph. However, the conclusion the authors draw is that "vehicle emissions are one of the major pollution sources in Beijing". My confusions regarding this paragraph are listed below. (a) Since Ph can be from both fossil fuel burning and vehicle emissions, what's the evidence to suggest that vehicles emissions contribute more to Ph than fossil fuel burning? (b) Do the authors suggest that Ph is mainly primary or secondary? Please be clear about vehicle emissions (i.e., primary) vs. the oxidation of vehicle emissions (i.e., secondary).

(3) Line 321 – 325. The first sentence discussed that $C_2$/Tot is the lowest in winter, indicating that organic aerosol in winter is less aged. This makes sense. However, the next sentence is "Because PM2.5 particles mainly originate from motor vehicles, fossil fuel and biomass combustion activities from local regions in winter, the aging process might occur during atmospheric transport." I can't see the link between these two sentences. What is the reason to mention "the aging process might occur during atmospheric transport?" Also, the authors need to support the statement that "$PM_{2.5}$ particles in winter are mainly primary", as many studies suggest that a large fraction of $PM_{2.5}$ is secondary in winter (Huang et al., 2014).

(4) Still in this paragraph. Line 326-330. The authors firstly compared the C2/Tot between Beijing and Central Himalayas. Then, the next sentence is the conclusion that "the photochemical

formation of dicarboxylic acids is insignificant in urban Beijing". How would this comparison justify the conclusion? What's the rationale to compare Beijing to Central Himalayas? The conclusion is over-stated to me. The only major evidence that authors provide in this paragraph is that C2/Tot is the lowest in winter than other seasons. This evidence can only suggest that OA is less aged and primary emissions contribute more to C2/Tot in winter than other seasons. It can't suggest whether dicarboxylic acids are mainly from primary or secondary. Also, it is inappropriate to use the C2/Tot ratio (i.e., only one dicarboxylic acids) to represent all dicarboxylic acids.

(5) Line 350-355. What's the rationale to correlate C2/wC2 to C2/Tot? What does the correlation mean? Please elaborate on what the negative correlation between C2/wC2 and C2/Tot suggest? It is not clear why these phenomena suggest these species are from biogenic burning emissions.

(6) Line 400. Missing connection between evidence and conclusion. "The outcomes above…" are merely some trends.

(7) Line 405-407. The Ph/C6 value is larger than unity for both diesel and gasoline. Thus, "Ph/C6 values larger than unity during the whole sampling year" can't justify that "diesel contributes more to diacids". The right evidence to imply the conclusion is that Ph/C6 values are closer to diesel than gasoline.

(8) Line 519-524. In the evidence, the authors compare the $\delta^{13}$C value between continental higher plants and marine plankton activities. However, the conclusion is that "anthropogenic primary emissions are important". I can't find the link between evidence and conclusion.

Also, Is C9 from anthropogenic emissions or the oxidation of anthropogenic emissions? Be clear.

(9) Line 534-535. Please provide evidence that tPh is related to plastic waste burning. For example, what is the $\delta^{13}$C value of tPh from plastic waster burning? I also want to point out that previous discussion in the manuscript (Line 269-271) does not provide evidence that tPh is related to plastic water burning. It is only mentioned that plastic water burning could be a source of tPh.

3.    Be consistent with terminology. For example, are "biomass combustion activities" (Line 324) the same as "biogenic burning" Line 354. what about "fossil fuel combustion" (Line 382) vs. "automobile emission" (Line 265) vs. "vehicle emission" (Line 268)?

4.    The PCA analysis does not provide further insights about the sources. The manuscript would benefit from more thorough evaluation of PCA results. For example, do the two factors represent different sources?

Minor Comments

1.      Line 36. It seems like that a word is missing after "relatively".

2.      Line 57-60. This sentence is confusing. The link between "WSOC/OC", "fraction of total carbon mass in particles", and "incomplete combustion activities" is unclear.

3.      Line 281-282. The sentence is not clear. Please rephrase.

4.      Line 291-292. Please show the R2 value when discussing the correlations.

5.      Line 311-316. These sentences just repeat Line 302-302 and don't offer any insights regarding the sources of Pyr and wC2.

6.      Line 337-339. It is not clear to me how this conclusion can be drawn.

7.      Line 520. What is the reason to compare C9 with C2-C4?

Reference

Huang, R.-J., Zhang, Y., Bozzetti, C., Ho, K.-F., Cao, J.-J., Han, Y., Daellenbach, K. R., Slowik, J. G., Platt, S. M., Canonaco, F., Zotter, P., Wolf, R., Pieber, S. M., Bruns, E. A., Crippa, M., Ciarelli, G., Piazzalunga, A., Schwikowski, M., Abbaszade, G., Schnelle-Kreis, J., Zimmermann, R., An, Z., Szidat, S., Baltensperger, U., Haddad, I. E., and Prevot, A. S. H.: High secondary aerosol contribution to particulate pollution during haze events in China, Nature, 514, 218-222, 10.1038/nature13774
http://www.nature.com/nature/journal/v514/n7521/abs/nature13774.html#supplementary-information, 2014.

---

## Referee Comment (RC2) · Anonymous Referee #1 · 23 Aug 2017

The manuscript presented the chemical characterization of a set of organics in PM2.5 from Beijing with information on compound-specific stable carbon isotopic ratios. The source identification or apportionment for particulate matters is a challenge task especially in highly polluted areas with complex primary and secondary sources. This study provided a year-round molecular distribution of organics with delta13C information. Detailed discussion was presented on the concentrations, ratios, and correlations among the individual compounds and total WSOC. The authors concluded that primary emissions such as biomass burning, fossil fuel combustion, and plastic burning, are the

major contributors to the organic acids and carbonyls. It is also concluded that the photochemical formation of these species in Beijing is insignificant. This study provides a set of valuable data on the particle phase organics, especially the compound-specific delta13C. The compound-specific delta13C data are useful for the source identification and may have other implications atmospheric chemistry. This is valuable for publication. However, the discussion and the statements in the current form can be further improved. Please see the following comments which the authors may need to consider in the revision.

Comments:

1, P4, L89, It is suggested the authors to provide a bit more background on the implication of stable carbon isotope ratios in atmospheric chemistry. The discussion and analysis of the compound-specific delta13C data can be further elaborated and compared to those at different geological locations if there is any.

2, P5, L129, What was the sampling time for each sample? How the sample was handled before analysis, this is critical for delta13C measurements?

3, P6, L 146, The manuscript should provide more details on the method of compound-specific delta13C, at least should be included in the supplementary. Current description on the method and quality control is over simplified.

4, P7, L173, The manuscript provided the backward trajectories, but there is only one sentence really discussed these information on P21, L578. Please also indicate in the caption of Fig. 1 the meanings of the numbers and colored backward trajectories.

5. Please be consistent with terminologiesÂăand abbreviations. The abbreviations are switched back and forth, such as C2/Oxalic acid, C4/Malonic acid. The terminologies sometimes are confusing including the vehicular/vehicle emissions, biomass burning activities/biogenic burning emissions, automobile emission, motor exhaust. If there is a difference between two similar ones, please define first to avoid the confusion.

6, Some statements or conclusions are not well justified. Here is a list of statements that the authors may need to elaborate or use different wording.

(1) P10, L267-268, the Ph concentration was higher in winter as compared to summer, but the contribution to the PM2.5 may be low. Also, how about the contribution of Ph from the photochemical oxidation and other fossil fuel burning other than vehicle emissions?

(2) P12, L330-331, The value of C2/Tot is compared to the case of Central Himalayas, why this particular location is chosen, how about the other locations? How do you evaluate the oxidation capability between these two locations which will certainly affect the C2/Tot ratios?

(3) P13, L345-354, the discussion in this paragraph is hard to follow. Simply base on the relationships among these species and drawing this conclusion (Line 353-354) is not convincing.

(4) P15, L405, The Ph/C6 ratio was lower than 2.5 in other seasons and it is only 4.1 in winter, but why it is concluded that more emissions from diesel burning (ratio of 6.58) than gasoline fuel vehicles (2.05)?

7, P20, L454, It is not clear what is connection between the information in L545-547 and the statement in L547-549.

8, It is suggested to use the words of "significant" or "insignificant" in the statements carefully unless statistical data or solid evidence are provided.

9, As presented in the manuscript, if I understood it correctly, the authors concluded that primary emissions are the major contributors to the organic acids and carbonyls. It is also concluded that the photochemical formation of these species in Beijing is insignificant. Both of these two statements are very strong. Is there any source apportionment study in Beijing during the same period, are they consistent or controversial?

10, Figure 6, please describe the meanings of different symbols and percentages for

the box. It is suggested to use same scale for different seasons.

---

## Referee Comment (RC3) · Anonymous Referee #3 · 25 Aug 2017

ACP review by Annonymous referee 1:

**MS title:** Molecular distribution and compound-specific stable carbon isotopic composition of dicarboxylic acids, oxocarboxylic acids, and α-dicarbonyls in PM2.5 from Beijing, China

Organic aerosols (OAs) account for major fraction of atmospheric particulate matter and also ubiquitous in nature. Among the OAs, the dicarboxylic acid and related polar compounds are one such widely studied chemical species that provide useful information about the relative significance of anthropogenic versus natural source contributions as well as primary emissions vs. secondary formation processes. In this context, combining the molecular distributions, concentrations, diagnostic mass ratios, air mass back trajectories and their stable carbon isotopic composition from this kind of studies are helpful in improving our current understanding of the complex nature of OAs. Therefore, the study is most relevant and publishable in ACP after a major revision.

I feel that conclusions are more clear and focussed than the most of the text part of this MS. The comparison of mass ratios among seasons are too vague. This should be supported by the statistical analysis such as ANOVA. To me, comparison of mean and sd of mass ratios of dicarboxylic acids among seasons appear to be insignificant for this study. In order to truly appreciate the relative significance of various source emissions (biogenic vs. anthropogenic) based on mass ratios of dicarboxylic acids and other related polar compounds, I strongly recommend the authors to evaluate their seasonal datasets using a statistical test (e.g, ANOVA). I see that there is a missing link in terms of attributing the stable carbon isotopic composition of dicarboxylic acids' with source contributions. For example, how the lowest $\delta^{13}C$ value of terephthalic acid (~-33.5‰) in winter indicates that it is emitted from plastic waste burning. Why not in other seasons? Is the plastic waste burning over Beijing is common only winter?

Another important issue is Section 3.6, 3.7 and 3.8: comparison of $\delta^{13}C$ of diacids and other compounds measured here makes this study unique due to year round sampling and comparing seasons. However, all these sections are bit complicated to follow/read. Since the East Asian outflow influences the chemical composition of organic aerosols during winter and spring, it is relevant to compare the diacid $\delta^{13}C$ values from this study with other sites/studies during this period only. No need to include autumn and summer. Therefore, I suggest authors to combine the winter

and spring data sets and use the median values to compare with the other sites in E. Asia (e.g. Sapporo, Gosan & cruises).

My other comments are as follows:

Line 35-37: The sentence is difficult to follow. Please rewrite.

Line 38-40: Correlations of some oxocarboxylic acids and a-dicarbonyls with nss-K+, how significant these are? Mention clearly what is correlated with what? Some oxocarboxylic acids are not specific!

Line 188-190: What is the reason that oxalic acid concentration is found to be highest in autumn and lowest in winter? Similarly, What causes the difference in the seasonality for the relative abundances of oxalic acid? Why is it maximum in summer? Explain/suggest.

Line 192: Update the references with a recent review.

Line 195 to 197. Authors need to provide, why there exist differences in the molecular distribution of measured water-soluble organic compounds among seasons and why do they show different patterns for e.g., why the third most abundant compound is glyoxylic acid in cold period and malonic acid in warmer period?.

Line 199-202: The sentence is not clear. What is single dicarboxylic acid?

Line 203: I am confused with subheading seasonal variability. Authors have already mentioned about differences in the molecular distributions of dicarboxylic acid among seasons already in the previous section. This section has to combine with the section 3.1.

Line 209: abbreviate $C_9$

Line 204-213: The seasonal trends were attributed to different emissions. This is not enough. Explain what source emissions might contribute for each type and also justify why you think this is the only possibility?

Line 214: Why the total diacid concentrations are the highest in autumn and why it is lowest in winter? Explain.

Line 216: Why Beijing dataset has to be compared with Tanzania, Africa? Both are different settings? Compare with polluted atmosphere with another city in S. Asia or E. Asia. Given the

diverse geographical locations, comparison with only one or two sites cannot be acceptable. Please compare or provide a table and discuss how different or similar this study site with those documented from other cities in China and India.

Line 223-230: After discussing the sources of oxalic acid, why there is a sudden jump to malonic acid data from this study. What about oxalic acid? If it is not important why authors are describing so much about its sources here. Connect here with their formation in different seasons. What are the different sources of oxalic acid, causing this variability through sampling period?

Line 231: Why malonic acid is highest in autumn?

Line 231-234: The connectivity between lines or sentences is missing. Why suddenly succinic acid to malonic acid ratio after mentioning the seasonal variability of malonic acid? What about the seasonal variability of succinic acid? Instead of picking up each compound measured and discussing its seasonality, I suggest authors to briefly summarize or infer logically the possible formation pathways of observed abundant compounds.

Line 231-235: Authors attributed the relative dominance of succinic acid over malonic acid as the major contribution from primary emissions to dicarboxylic acids measured here. Although this could be possible, however, one cannot rule about the transport during each season. So if you see the air mass back trajectories at the receptor site, then this inference based on C4/C3 has certain uncertainty or bias. So you need to mention this in the MS.

Line 235: The diurnal variation tendency of C2?? Is it diurnal or daily variability?

Line 255-257: Why the correlation of azelaic acid (C9) with K+ and Cl- solely attributed to coal burning? Why not biomass burning?

Line 269-271: According to authors "The predominance of terephthalic acid over phthalic acid observed in this study is in contrast with those reported by Ho et al., ". Is it due to variability in the sources or increase plastic waste burning is increasing. Comment on this.

Line 277-278: Provide a reference for the argument that monocarboxylic acids are photochemically oxidized & form dicarboxylic acid. Why authors think it is relevant here rather than direct emissions or other sources.

Line 281: the sentence is not clear.

Line 290–292: These sentences are not clear. You can combine into one as "$\omega C_2$ and Pyr is more abundant in cold seasons (Table 1) and correlated with $K^+$ and $Cl^-$". What is the common combustion source, mention it?

Line 326-328: I don't follow the comparison with the Central Himalayan aerosols as well as the logic of the statement. rewrite.

Line 333-339: In urban Beijing, how can authors assume that succinic acid formation forms the photooxidation of unsaturated fatty acids? What about the photochemical oxidation of adipic acid, which is a product of cyclic olefins with oxidants in and around the city? Why not is C4 derived from anthropogenic emissions such as fossil fuel combustion, vehicular emissions in Beijing? The linear relationship between C2/Tot (or relative abundance of oxalic acid in total diacid mass) and the C2/C4 just indicate that oxalic acid has a significant contribution from or formed from the photochemical oxidation of succinic acid, not more than that. So authors need to dilute their emphasis on source attribution directly based on a linear relationship. If still, authors think that succinic acid might have produced from the photochemical breakdown of higher homologues of dicarboxylic acids from the biogenic unsaturated fatty acids, they should the linear relationships with oleic acid and azelaic acid first and then lower homologues of dicarboxylic acids with azelaic acid. I suggest authors think and rewrite along these lines.

Line 335: Authors need to provide a proper reference for invoking contribution of emissions from the phytoplankton in remote oceans (update the reference with diacid review, doi: 10.1016/j.atmosres.2015.11.018 and others cited in).

Line 340-344 and 345-346 is missing. Connect these two as "We, therefore, would like to investigate the significance of these formation pathways by examining the interrelationships between…"

Line 347: correct the sentence as "in all seasons except winter"

Line 352-353: how the negative correlation between oxalic to glyoxylic acid mass ratio and the relative abundance of oxalic acid in summer demonstrate that $C_2$, $\omega C_2$, Pyr and Gly are from biomass burning emissions? Elaborate further.

Line 361: provide a reference that $C_3/C_4$ is a good indicator for dicarboxylic acid contribution from primary emissions vs. secondary formation process.

Line 366: correct the sentence as "throughout the sampling year"

Line 367-368. I couldn't find any difference between the two statements.

Line 376-377: Confusing!. State clearly that the direct emissions from localized sources in Beijing contributed to atmospheric dicarboxylic acids.

Line 380-384: I understand for the summer season that Ph might have a contribution from photochemical oxidation of PAHs. What is the reason for its higher abundance in winter season?

Line 396-398: As stated in Line 385, if both adipic acid (C6) and phthalic acid (Ph) are produced from the photochemical oxidation of cyclic olefins, why C6/C9 is lowest in winter and whereas Ph/C9 is highest in winter. Perhaps both could have been produced by different sources. That is the reason why comparing seasonal means directly can yield erroneous interpretations. May be the seasonal averages are not significantly different.

Line 410-414: I am not able to follow the comparison of ratios. Authors need to evaluate the differences in the seasonal means using ANOVA.

Why not authors use PMF other than PCA for the source apportionment in this study, like their previous publications?

Section 3,4: I do not understand the title of this subheading. Already authors made a comparison in the previous section. Then why suddenly there is another section on this?

Line 519-523: I do not follow the logic here. How authors drew their conclusion that d13C of C9 indicate that azelaic acid is from biomass burning in surrounding areas?

Line 534-535: How the lowest $\delta^{13}C$ value of terephthalic acid (~-33.5‰) indicates that it is emitted from plastic waste burning that too in winter. Why not in other seasons? Is the plastic waste burning over Beijing is common only winter. Line 534-535 and the next sentence are not connected. The reference cited is the work related to S. Asia. How about the conditions in E. Asia? Is the plastic waste burning is relevant for the study site? If so, please state it.

Line 538: What major compounds? Be specific (diacids, oxoacids or a-dicarbonyls or what you are referring to).

Line 540: The decreasing trend in d$^{13}$C of C5 to C9 is not obvious from the Figure 6 for all seasons (see for e.g., summer, panel d).

Line 542-544: given the overlap of the box widths for oxalic, malonic and succinic acids, I wonder whether the 13C enrichment for these compounds is significantly different among seasons.

The scale in Figure 6 is most difficult to compare between seasons. I suggest authors keep the same scale and then discuss the differences.

Figure 8: I do not follow the caption. What are the concentrations plotted on the x-axis.

---

## Author Comment (AC1) · 2 Nov 2017

Responses to the Comments of Referee #2

Review of "Molecular distribution and compound-specific stable carbon isotopic composition of dicarboxylic acids, oxocarboxylic acids, and α-dicarbonyls in PM2.5 from Beijing, China" by Zhao et al.

In this study, the authors measured the concentrations and stable carbon isotopic composition of dicarboxylic acids, oxocarboxylic acids, and α-dicarbonyls in Beijing, China. The annual variation of species, mass ratios between species, and δ13C are used to infer the sources and formation processes of measured species. The authors found that (1) anthropogenic primary emissions are the major contribution to the measured species and (2) the photooxidation is weak and contributes insignificant amount of measured species. However, I find the conclusions are not well justified. Particularly, many discussions suffer flaws in its logical flow and it is unclear how the conclusions are drawn from the analysis/evidence. also, some discussions contradict with each other. Therefore, I would not recommend this manuscript for publication in its current state.

Response: We thank the referee's careful reading and helpful comments, based on which we tried to improve the quality of our manuscript as follows.

Major Comments

1. Many discussions contradict with each other. This causes severe confusion and leaves the conclusions ambiguous. One major conclusion in this study is that "the degree of photochemical formation of diacids in Beijing is insignificant". However, there are many discussions in the manuscript which contradict with this conclusion. For example, Line 337-339. The authors state that "C2/Tot showed strong correlations with C2/C4 in all four seasons, indicating the significance of photooxidation pathways of biogenic unsaturated fatty acids". This sentence directly contradicts the main conclusion of this study.

Response: Thanks for your suggestion. We have rephrased the sentence in the revised MS. "Typically, concentrations of C2, C4 and C9 diacids showed similar seasonal variation trends (Fig. 1), implying that they were derived from common primary emissions or

photochemical processing. Furthermore, C2/Tot (C2%) showed strong correlations with C2/C4 in all four seasons (Fig. S4), indicating the importance of biogenic unsaturated fatty acids, followed by secondary formation ways." Please see line 344–348 in page 13.

In Line 419-421, it is stated that "the high ratios of M/F throughout the whole year imply that aerosols in Beijing are not seriously subjected to secondary oxidation process". While the photooxidation may be weak in winter, it is strong in summer. The statement in Line 419-421 directly contradicts to many studies (for example, those cited in the introduction section). How do authors reconcile the discrepancies?

Response: We have added the sentence as "Though M/F obtained the lowest ratio values in summer, dicarboxylic acids and related compounds in Beijing are not seriously subjected to secondary oxidation process in the whole year with comparison to the strength of primary emissions." Please see line 432–435 in the revised MS.

Many studies cited in the introduction prefer to study the formation mechanisms and source apportionments of aerosols in haze days, while we focused more on the characterization of water-soluble organic acids in Beijing at a molecular level. Different sampling time and study target may get different conclusions.

Zhang et al. (2013) found that soil dust, coal combustion, biomass burning, traffic and waste incineration emission, industrial pollution, and secondary inorganic aerosol (SIA) are six main sources for PM2.5 in Beijing. Each of these sources has an annual mean contribution of 16, 14, 13, 3, 28, and 26 %, respectively, to PM2.5. Similarly, the relative contributions of these sources to PM2.5 in Beijing greatly varied with the changing seasons, which proves the "complex air pollution" in Beijing. The highest contributions occurred in spring for soil dust (23 %) and traffic and waste incineration emission (5 %), in winter for coal combustion (57 %), in autumn for industrial pollution (42 %), in summer for SIA (54 %), and in both spring and autumn for biomass burning (19 and 17 %, respectively). The conclusions in this article demonstrated that regional primary sources could be crucial contributors to PM pollution in Beijing, compared to the enhanced photochemical formations of SIA in summer.

Some other contradictions are listed below. This is by no means an exclusive list. The authors must check carefully about inconsistency throughout the manuscript.

In Line 381, "these findings imply that Ph is largely emitted by anthropogenic sources in winter". In line 385, "Ph are mostly formed via secondary oxidations of anthropogenic aromatic hydrocarbons."

Response: We have rephrased these sentences in in the revised MS. "These findings imply that phthalic acid is largely emitted by anthropogenic sources in winter, mainly as a result of intensive fossil fuel combustion. It is worth noting Ph/Tot ratio was also relatively high in summer. Previous study reported that a great amount of naphthalene obtained in Beijing is an important raw material for the substantial formation of phthalic acid (Liu et al., 2007). Therefore increased ambient temperatures and stronger solar radiation in summertime facilitate the transformation of gaseous PAHs (e.g., naphthalene) to produce relative high levels of Ph in Beijing.

C6 and Ph can be formed via secondary oxidations of anthropogenic cyclic olefins (e.g., cyclohexene) and aromatic hydrocarbons, respectively, whereas C9 is mainly produced by photochemical oxidation of biogenic unsaturated fatty acids (Kawamura and Gagosian, 1987;Kawamura and Ikushima, 1993)." Anthropogenic sources include biomass burning, fossil fuel burning and plastic waste burning, motor vehicles. Please see line 391–401.

In Line 239-242 and Line 257, "...indicating that substantial amounts of C9 may be stemmed from the local and surrounding combustion activities in Beijing." Line 386, "C9 is a photochemical product of biogenic unsaturated fatty acids." Then, C6/C9 is used to evaluate the source strength of anthropogenic vs biogenic emissions.

Response: Biomass burning may be technically defined as anthropogenic combustion activity (at least this is the case for 90% of all fires on a global basis), but the materials are mostly of biogenic origin. Hence, azelaic acid could be produced by the biomass burning because the biomass should contain its precursors (unsaturated fatty acids)(Kawamura et al., 2013).

Similarly, during the biomass burning, cyclic olefins may also be produced in a similar manner as in fossil fuel combustion processes and further oxidized to C6 in the

atmosphere during the transport. Although these possible pathways may cause a slightly complicated situations of C6/C9 ratios as a tracer, previous study reported that contributions of C6 and Ph from biomass burning are minimal (Kawamura et al., 2013). Therefore, C6/C9 is used to evaluate the source strength of anthropogenic vs biogenic emissions.

2. Many discussions suffer flaws in logical flow and the link between evidence and conclusion is not clear. To name a few,

(1) Line 43-46. Why could "lower δ13C of major species in Beijing than western North Pacific" indicate "weak photooxidation in Beijng"? What is the rationale to compare Beijing with western North Pacific?

Response: "The systematic differences in stable carbon isotope ratios of diacids and other polar acids were attributable to kinetic isotope fractionation processes in the atmosphere (Hoefs and Hoefs, 1997), while secondary oxidation of these water-soluble organic acids is more influential for diacid carbons to enrich in 13C (Wang and Kawamura, 2006). For example, the relatively short carbon-chain diacids enriched in 13C were ascribed to the kinetic isotopic effect (KIE) for the photochemical breakdown of longer-chain diacids (Anderson et al., 2004;Irei et al., 2006). And lower dicarboxylic acids with enrichment of 13C may be less active to oxidants (e.g., OH radicals). Therefore, the determinations of δ13C values of dicarboxylic acids and related compounds show vital information about the atmospheric aging processes of aerosols derived from local emissions or long-range transport ways in air." Please see line 518–527 in the revised MS.

Particulate organic matters in remote marine areas are intensively aged during long-range transport and are affected by both the sea-to-air emissions and the terrestrial outflows (Wang and Kawamura, 2006). In general, δ13C values of diacids from marine aerosols are larger than those obtained in Beijing aerosols. This enrichment in 13C can be explained by the isotopic fractionation for more aged aerosols. Please see line 591–595 in the revised MS.

(2) Line 258-268. Firstly, the authors mentioned that phthalic acids (Ph) can be formed

vis photooxidation of naphthalene or directly emitted by fossil fuel burning and incomplete combustion. Secondly, the authors stated that the great amount of naphthalene is the precursor of Ph. However, the conclusion the authors draw is that "vehicle emissions are one of the major pollution sources in Beijing". My confusions regarding this paragraph are listed below. (a) Since Ph can be from both fossil fuel burning and vehicle emissions, what's the evidence to suggest that vehicles emissions contribute more to Ph than fossil fuel burning? (b) Do the authors suggest that Ph is mainly primary or secondary? Please be clear about vehicle emissions (i.e., primary) vs. the oxidation of vehicle emissions (i.e., secondary).

Response: We have rephrased these sentences in in the revised MS. "Phthalic acid is either formed via photochemical pathways of naphthalene or directly released into air by fossil fuel burning and the incomplete combustion of aromatic hydrocarbons in motor vehicles. Moreover, the abundance of Ph may also be caused by increased phthalates emissions from plastic waste burnings in heavily polluted areas in China (Deshmukh et al., 2015). Phthalic acid esters are used as plasticizers in resins and polymers (Simoneit et al., 2005). Therefore, anthropogenic emissions contributed to relatively high concentrations of Ph in PM2.5 in Beijing." Please see line 268–274 in the revised MS.
No matter whether phthalic acid is formed via the secondary oxidation ways of precursors, the emission sources are defined as anthropogenic activities, such as plastic waste burning, biomass burning, fossil fuel burning as well as motor vehicles.

(3) Line 321–325. The first sentence discussed that C2/Tot is the lowest in winter, indicating that organic aerosol in winter is less aged. This makes sense. However, the next sentence is "Because PM2.5 particles mainly originate from motor vehicles, fossil fuel and biomass combustion activities from local regions in winter, the aging process might occur during atmospheric transport." I can't see the link between these two sentences. What is the reason to mention "the aging process might occur during atmospheric transport?" Also, the authors need to support the statement that "PM2.5 particles in winter are mainly primary", as many studies suggest that a large fraction of PM2.5 is secondary in winter (Huang et al., 2014).

Response: We have rephrased these sentences in in the revised MS. "As for water-soluble organic acids in this paper, the ratios of $C_2$/Tot were the lowest in winter ($0.39\pm0.05$) (Table 3), indicating that wintertime organic aerosols may be less aged (Fig. 4a). After the main emission of $PM_{2.5}$ particles from motor vehicles, fossil fuel and biomass burning activities from local regions in winter, the aging process might occur during atmospheric transport." Please see line 328–332 in the revised MS. Huang et al. (2014) concluded that in comparison with secondary sources, primary emissions contributed slightly less to fine particles in haze events at urban locations in China. Their study focused on the fine aerosol and used both molecular markers and radiocarbon (14C) measurements. Different sampling time and locations are related with various strength of primary emissions and atmospheric conditions. Many scholars like to study the formation mechanisms and source apportionments of aerosols in haze days, while we focused more on the characterization of water-soluble organic acids in common days containing polluted and clean days in Beijing at a molecular level. Different sampling time and study target may get different conclusions.

(4) Still in this paragraph. Line 326-330. The authors firstly compared the C2/Tot between Beijing and Central Himalayas. Then, the next sentence is the conclusion that "the photochemical formation of dicarboxylic acids is insignificant in urban Beijing". How would this comparison justify the conclusion? What's the rationale to compare Beijing to Central Himalayas? The conclusion is over-stated to me. The only major evidence that authors provide in this paragraph is that C2/Tot is the lowest in winter than other seasons. This evidence can only suggest that OA is less aged and primary emissions contribute more to C2/Tot in winter than other seasons. It can't suggest whether dicarboxylic acids are mainly from primary or secondary. Also, it is inappropriate to use the C2/Tot ratio (i.e., only one dicarboxylic acids) to represent all dicarboxylic acids.

Response: The ratio of C2/total dicarboxylic acids can be used to assess the aging process of organic aerosols, because oxalic acid has been recognized as the end product that is associated with atmospheric chain reactions of organic species with oxidants. Typically, higher ratios are observed with the progress of aerosol aging (Kawamura and Sakaguchi,

1999).

Over south Asia, Indo-Gangetic plain is considered as a densely populated region, and thus as a potentially strong source region of anthropogenic aerosols. Northern part of these highly populated and industrialized areas is one of chains of Himalaya Mountains. Due to its high elevation, the Himalayan range acts as a boundary limiting the northern extent of the Indian summer monsoon, and therefore, observations at a high altitude location, Nainital (29.4° N; 79.5° E, 1958 m a.s.l.) would provide information about emissions over the Indian subcontinent. Their observation site is located at the highest mountain top (over Kumaon region) and about 2 km far from Nainital city (population ~0.5 million). The site is devoid of any major local pollution sources nearby and is generally free from the snow coverage during most of the time. North and northeast side of the study area are characterized by sharply peaking topography of Himalayan mountain ranges, whereas south-western side plains with very low elevation (<500 m a.s.l.) are densely populated with land merging into the Ganga basin (Hegde and Kawamura, 2012).

The C2/total diacid ratios show higher values in winter (˜0.8±0.04) than summer (˜0.5±0.01), suggesting that the winter aerosols may be more aged in Central Himalayas. As the anthropogenic aerosols that are emitted from the industrial regions of Indo Gangetic Plain areas can travel to the north and reach the sampling site by the northerly wind (comparatively lower temperature and weaker wind speed) during the winter period, aging of these aerosols might occur during the transport and thereby significantly contribute to the higher C2/total diacid ratios. In contrast, this trend is reversed during summer. Because the temperature over the region increase and high wind favors quick transport of pollutants, fresher aerosols are transported over the sampling site (Hegde and Kawamura, 2012).

The C2/Tot ratios were the lowest in winter (0.39±0.05), while C2/Tot ratios are similar in the other three seasons (0.54–0.58). So it's rational to compare the values of C2/total diacid ratio in Beijing with those in Central Himalayas to find out that the photochemical formation of dicarboxylic acids is insignificant in urban Beijing. Please see line 328–332 in the revised MS.

(5) Line 350-355. What's the rationale to correlate C2/wC2 to C2/Tot? What does the correlation mean? Please elaborate on what the negative correlation between C2/wC2 and C2/Tot suggest? It is not clear why these phenomena suggest these species are from biogenic burning emissions.

Response: 1. Please see line 348–357 in the revised MS, where elaborated the meaning of correlation between C2/$\omega$C$_2$ and C2/Tot.

2. The negative correlation between C2/$\omega$C2 and C2/Tot suggests that supplement of $\omega$C2 was faster than their secondary transformations to C2 in Beijing. The aging level was not strong enough for intermediate diacids contributing to the production of C2.

3. Please see line 297–300 in the revised MS and figure S2 in supporting material, where stated these species are from biogenic burning emissions. Potassium ion (K+) is a good tracer of biomass burnings (Andreae, 1983). And a great deal of chloride in wintertime Beijing is linked to increased emissions of coal incineration, particularly under stagnant meteorological conditions that facilitate the formation of particle-phase ammonium chloride (Sun et al., 2013). Please see line 256–260 in the revised MS, where stated the meanings of K$^+$ and Cl$^-$.

(6) Line 400. Missing connection between evidence and conclusion. "The outcomes above..." are merely some trends.

Response: The mass ratios of diacids in comparison with those in other sampling sites can better assess the emission strength of anthropogenic activities in Beijing.

Please see line 407–414 in the revised MS.

(7) Line 405-407. The Ph/C6 value is larger than unity for both diesel and gasoline. Thus, "Ph/C6 values larger than unity during the whole sampling year" can't justify that "diesel contributes more to diacids". The right evidence to imply the conclusion is that Ph/C6 values are closer to diesel than gasoline.

Response: The Ph/C6 ratio were comparable to or lower than 2.5 in other seasons and only showed the largest mean value (4.1) in winter. Following the reviewer' suggestion,

we have revised the text as follows.

"Ph/C6 ratios reached to the highest values in winter (4.06±0.78) and the lowest in autumn (1.66±0.78). Kawamura and Kaplan (1987) demonstrated that the Ph/C6 ratio from gasoline fuel vehicle (2.05) is lower than that from diesel fuel vehicles (6.58). This phenomena shows abundances of diacids attributable to more emissions from gasoline fuel vehicles than diesel burning."

Please see line 416–420 in the revised MS.

(8) Line 519-524. In the evidence, the authors compare the δ13C value between continental higher plants and marine plankton activities. However, the conclusion is that "anthropogenic primary emissions are important". I can't find the link between evidence and conclusion.

Also, Is C9 from anthropogenic emissions or the oxidation of anthropogenic emissions? Be clear.

Response: Rephrased. Thanks for your suggestion. Please see line 536–541 in page 20.

(9) Line 534-535. Please provide evidence that tPh is related to plastic waste burning. For example, what is the δ13C value of tPh from plastic waster burning? I also want to point out that previous discussion in the manuscript (Line 269-271) does not provide evidence that tPh is related to plastic water burning. It is only mentioned that plastic water burning could be a source of tPh.

Response: In general, secondary oxidation of water-soluble organic acids is more influential for diacid carbons to enrich in 13C (Wang and Kawamura, 2006). tPh is a good tracer of open-waste burning (Kumar et al., 2015). Therefore, the low δ13C values of tPh in winter indicated that tPh is directly emitted plastic waste burning.

The result given by Ho et al. (2010) in line 276–277 reported Ph as the second most abundant compound in Beijing. The purpose of Ho's result exhibited in the revised MS is as a contrast with our conclusion to illustrate the abundance of tPh in Beijing. Simoneit et al. (2005) provided evidence that tPh is related to plastic waste burnings. It has been stated in line 277–278 in the revised MS.

3. Be consistent with terminology. For example, are "biomass combustion activities" (Line 324) the same as "biogenic burning" Line 354. what about "fossil fuel combustion" (Line 382) vs. "automobile emission" (Line 265) vs. "vehicle emission" (Line 268)?

Response: Corrected.

4. The PCA analysis does not provide further insights about the sources. The manuscript would benefit from more thorough evaluation of PCA results. For example, do the two factors represent different sources?

Response: The PCA analysis further showed solid evidence to prove the results found in previous sections.

Minor Comments

1. Line 36. It seems like that a word is missing after "relatively".

Response: Corrected. Thanks for your suggestion.

2. Line 57-60. This sentence is confusing. The link between "WSOC/OC", "fraction of total carbon mass in particles", and "incomplete combustion activities" is unclear.

Response: A major fraction of organic aerosols is water-soluble, accounting for 20−60% of aerosol carbon mass in fossil fuel combustion-derived particles and 45−75% of that in biomass burning-derived particles (Pathak et al., 2011;Falkovich et al., 2005).

3. Line 281-282. The sentence is not clear. Please rephrase.

Response: We have rephrased the sentence. Please see line 287–289 in the revised MS.

4. Line 291-292. Please show the R2 value when discussing the correlations.

Response: Please see figure S2 and figure S3 in the supporting material.

5. Line 311-316. These sentences just repeat Line 302-302 and don't offer any insights regarding the sources of Pyr and wC2.

Response: The line 309–313 is the conclusions reported in previous studies, while the line 315–323 is the demonstration of sources in our paper.

6. Line 337-339. It is not clear to me how this conclusion can be drawn.

Response: In figure 3, a big pollution process was observed from 10.3 to 10.8 in autumn, followed by secondary oxidation ways, which leaded to the increased concentrations of organic acids in fall. C2, C4, C9, ωC2 and Pyr showed similar concentration variations, indicating they may have common primary sources or photochemical pathways. In section 3.2, C9, ωC2, Pyr, Gly and MeGly correlated well with nss-K+, further implying that these organic acids were emitted from biomass burnings.

We have rephrased "Typically, concentrations of C2, C4 and C9 diacids, as well as ωC2 and Pyr showed similar varational trends in autumn, implying that they were derived from similar primary emissions or photochemical processing. Furthermore, C2/Tot (C2%) showed strong correlations with C2/C4 in all four seasons (Fig. S4), indicating the importance of biogenic unsaturated fatty acids, followed by photochemical processing." Please see line 344–348 in the revised MS.

7. Line 520. What is the reason to compare C9 with C2-C4?

Response: "For example, the relatively short carbon-chain diacids enriched in 13C were ascribed to the kinetic isotopic effect (KIE) for the photochemical breakdown of longer-chain diacids (Anderson et al., 2004; Irei et al., 2006). And lower dicarboxylic acids with enrichment of 13C may be less active to oxidants (e.g., OH radicals). Therefore, the determinations of δ13C values of dicarboxylic acids and related compounds show vital information about the atmospheric aging processes of aerosols derived from local emissions or long-range transport ways in air." Please see line 521–527 in the revised MS.

The δ13C values of C9 in comparison with those of C2–C4 is to connect with the conclusions in previous studies.

References:

Anderson, R. S., Huang, L., Iannone, R., Thompson, A. E., and Rudolph, J.: Carbon kinetic isotope effects in the gas phase reactions of light alkanes and ethene with the OH radical at 296±4 K, The Journal of Physical Chemistry A, 108, 11537-11544, 2004.

Andreae, M. O.: Soot carbon and excess fine potassium: long-range transport of combustion-derived aerosols, Science, 220, 1148, 1983.

Falkovich, A., Graber, E., Schkolnik, G., Rudich, Y., Maenhaut, W., and Artaxo, P.: Low molecular weight organic acids in aerosol particles from Rondonia, Brazil, during the biomass-burning, transition and wet periods, Atmospheric Chemistry and Physics, 5, 781-797, 2005.

Hegde, P., and Kawamura, K.: Seasonal variations of water-soluble organic carbon, dicarboxylic acids, ketocarboxylic acids, and α-dicarbonyls in Central Himalayan aerosols, Atmospheric Chemistry and Physics, 12, 6645-6665, 2012.

Ho, K., Lee, S., Ho, S. S. H., Kawamura, K., Tachibana, E., Cheng, Y., and Zhu, T.: Dicarboxylic acids, ketocarboxylic acids, α-dicarbonyls, fatty acids, and benzoic acid in urban aerosols collected during the 2006 Campaign of Air Quality Research in Beijing (CAREBeijing-2006), Journal of geophysical research: atmospheres, 115, 2010.

Hoefs, J., and Hoefs, J.: Stable isotope geochemistry, Springer, 1997.

Irei, S., Huang, L., Collin, F., Zhang, W., Hastie, D., and Rudolph, J.: Flow reactor studies of the stable carbon isotope composition of secondary particulate organic matter generated by OH-radical-induced reactions of toluene, Atmospheric Environment, 40, 5858-5867, 2006.

Kawamura, K., and Sakaguchi, F.: Molecular distributions of water soluble dicarboxylic acids in marine aerosols over the Pacific Ocean including tropics, Journal of Geophysical Research: Atmospheres, 104, 3501-3509, 1999.

Kawamura, K., Tachibana, E., Okuzawa, K., Aggarwal, S., Kanaya, Y., and Wang, Z.: High abundances of water-soluble dicarboxylic acids, ketocarboxylic acids and α-dicarbonyls in the mountaintop aerosols over the North China Plain during wheat burning season, Atmospheric chemistry and physics, 13, 8285-8302, 2013.

Kumar, S., Aggarwal, S. G., Gupta, P. K., and Kawamura, K.: Investigation of the tracers for plastic-enriched waste burning aerosols, Atmospheric Environment, 108, 49-58, 2015.

Pathak, R. K., Wang, T., Ho, K., and Lee, S.: Characteristics of summertime PM 2.5 organic and elemental carbon in four major Chinese cities: implications of high acidity for water-soluble organic carbon (WSOC), Atmospheric Environment, 45, 318-325, 2011.

Simoneit, B. R., Medeiros, P. M., and Didyk, B. M.: Combustion products of plastics as indicators for refuse burning in the atmosphere, Environmental science & technology, 39, 6961-6970, 2005.

Sun, Y., Wang, Z., Fu, P., Yang, T., Jiang, Q., Dong, H., Li, J., and Jia, J.: Aerosol composition, sources and processes during wintertime in Beijing, China, Atmospheric Chemistry and Physics, 13, 4577-4592, 2013.

Wang, H., and Kawamura, K.: Stable carbon isotopic composition of low-molecular-weight dicarboxylic acids and ketoacids in remote marine aerosols, Journal of Geophysical Research: Atmospheres, 111, 2006.

Zhang, R., Jing, J., Tao, J., and Hsu, S. C.: Chemical characterization and source apportionment of PM2.5 in Beijing: seasonal perspective, Atmospheric Chemistry & Physics, 13, 7053-7074, 2013.

---

## Author Comment (AC2) · 2 Nov 2017

The manuscript presented the chemical characterization of a set of organics in PM2.5 from Beijing with information on compound-specific stable carbon isotopic ratios. The source identification or apportionment for particulate matters is a challenge task especially in highly polluted areas with complex primary and secondary sources. This study provided a year-round molecular distribution of organics with delta13C information. Detailed discussion was presented on the concentrations, ratios, and correlations among the individual compounds and total WSOC. The authors concluded that primary emissions such as biomass burning, fossil fuel combustion, and plastic burning, are the major contributors to the organic acids and carbonyls. It is also concluded that the pho-tochemical formation of these species in Beijing is insignificant. This study provides a set of valuable data on the particle phase organics, especially the compound-specific delta13C. The compound-specific delta13C data are useful for the source identification and may have other implications atmospheric chemistry. This is valuable for publica- tion. However, the discussion and the statements in the current form can be further improved. Please see the following comments which the authors may need to consider in the revision.

Response: We thank the reviewer's careful reading and important comments, which improve the quality of our manuscript.

Comments:

1, P4, L89, It is suggested the authors to provide a bit more background on the implication of stable carbon isotope ratios in atmospheric chemistry. The discussion and analysis of the compound-specific delta13C data can be further elaborated and compared

to those at different geological locations if there is any.

Response: As suggested by the reviewer, the following words have been added in the revised manuscript. Furthermore, the analyses of stable carbon isotope ratios of water-soluble organic acids can be effectively applied to assessing the photochemical aging level and relative contributions of primary emissions to aerosol samples in atmosphere using the estimated kinetic isotope effect of target compound with OH radical (Kawamura and Watanabe, 2004; Wang and Kawamura, 2006; Kawamura and Bikkina, 2016). Please see line 88–92 in page 4.

The discussion and analysis of the compound-specific delta13C data is further elaborated in section 3.6–3.8.

The δ13C values of diacids and related compounds in Beijing is compared with other areas in sections 3.7 in the revised MS.

2, P5, L129, What was the sampling time for each sample? How the sample was handled before analysis, this is critical for delta13C measurements?

Response: Thanks for your suggestion. Each sample was collected onto pre-heated (450°C, 6 hours) quartz-fiber filters (Pallflex) by using a high-volume air sampler (TISCH, USA) at an airflow rate of 1.0 m$^3$/min for 23 h from September 2013 to July 2014 (n=65). We have rephrased and added some sentences in the revised MS. Please see line 131–133 in page 5.

3, P6, L146, The manuscript should provide more details on the method of compound-specific delta13C, at least should be included in the supplementary. Current description on the method and quality control is over simplified.

Response: Thanks for your suggestion. We have added some sentences. "In short, some internal standard (n-alkane C13) was added into derivatized fraction of each sample at a proper proportion. δ13C values of the derivatized dibutyl esters or dibutoxy acetals, measured using GC (HP6890)/isotope ratio mass spectrometer (irMS), were then calculated for diacids, ketoacids and α-dicarbonyls based on the isotopic mass balance equation." Please see Page 6, Line 151–155 in the revised MS.

4, P7, L173, The manuscript provided the backward trajectories, but there is only one sentence really discussed these information on P21, L578. Please also indicate in the caption of Fig. 1 the meanings of the numbers and colored backward trajectories.

Response: Thanks for your suggestion. We have rephrased and added some sentences. "Burning activities in East Asia were illustrated by fire spots, and the datasets were downloaded from MODIS website (https://firms.modaps.eosdis.nasa.gov/download/request.php). The backward trajectories were assorted into several major classifications on the basis of prevalent winds direction. And the numbers in each panel imply the percentages of hourly trajectories in the sampling season to better illustrate the air mass origins, as given below in this study (Fig. 1)."

Please see Page 7, Line 181–186 and Page 42, Line 1000–1001 in the revised MS.

5. Please be consistent with terminologiesÂaˇand abbreviations. The abbreviations are switched back and forth, such as C2/Oxalic acid, C4/Malonic acid. The terminologies sometimes are confusing including the vehicular/vehicle emissions, biomass burning activities/biogenic burning emissions, automobile emission, motor exhaust. If there is a difference between two similar ones, please define first to avoid the confusion.

Response:

Reponse: We are sorry for this mistake. Biogenic burning emissions may be not same as biomass burning activities. We have corrected in the revised MS.

There is no difference between vehicular emissions, vehicular exhaust and motor vehicles.

6, Some statements or conclusions are not well justified. Here is a list of statements that the authors may need to elaborate or use different wording.

(1) P10, L267-268, the Ph concentration was higher in winter as compared to summer, but the contribution to the PM2.5 may be low. Also, how about the contribution of Ph from the photochemical oxidation and other fossil fuel burning other than vehicle emissions?

Response: Phthalic acid is either formed via photochemical pathways of naphthalene or directly released into air by fossil fuel burning and the incomplete combustion of aromatic hydrocarbons in motor vehicles. Moreover, the abundant presence of Ph may also be caused by enhanced emission of phthalates from plastics used in heavily populated and industrialized regions in China.

The relative abundance of Ph in total diacids was lower than the value of C2/Tot ratio. But concentrations of Ph and Ph/Tot both showed the highest values in winter. Combining the ratios of C2/Tot and C3/C4, the photochemical aging tracer of organic aerosol, which obtained the lowest values in winter and further indicated the substantial anthropogenic primary emissions, we found that photochemical oxidation pathways of Ph is insignificant compared to strength of primary emissions in wintertime. The highest concentration of Ph is attributable to the enhanced fossil fuel burning for house heating and stagnant atmospheric conditions during wintertime in Beijing. But Ph/Tot ratio was also relatively high in summer. Previous study reported that a great amount of naphthalene obtained in Beijing is an important raw material for the substantial formation of phthalic acid (Liu et al., 2007). Therefore increased ambient temperatures and stronger solar radiation in summertime facilitate the transformation of gaseous PAHs (e.g., naphthalene) to produce relative high levels of Ph in Beijing.

We have corrected the mistake and rephrased "Phthalic acid is either formed via photochemical pathways of naphthalene, or directly released into air by fossil fuel burning and the incomplete combustion of aromatic hydrocarbons in motor vehicles. Moreover, the abundance of Ph may also be caused by increased phthalates emissions from plastic waste burnings in heavily polluted areas in China (Deshmukh et al., 2015). Phthalic acid esters are used as plasticizers in resins and polymers (Simoneit et al., 2005). Therefore, anthropogenic activities contributed to relatively high concentrations of Ph in PM2.5 in Beijing." Please see line 268–274 in the revised MS.

"The Ph/Tot ratios in winter were nearly 2–3 times greater than those in spring and autumn. These findings imply that phthalic acid is largely emitted by anthropogenic sources in winter, mainly as a result of intensive fossil fuel combustion. It is worth noting that Ph/Tot ratio was also relatively high in summer. Previous study reported that a great

amount of naphthalene obtained in Beijing is an important raw material for the substantial formation of phthalic acid (Liu et al., 2007). Therefore increased ambient temperatures and stronger solar radiation in summertime facilitate the transformation of gaseous PAHs (e.g., naphthalene) to produce relative high levels of Ph in Beijing." Please see line 390–397 in the revised MS.

(2) P12, L330-331, The value of C2/Tot is compared to the case of Central Himalayas, why this particular location is chosen, how about the other locations? How do you evaluate the oxidation capability between these two locations which will certainly affect the C2/Tot ratios?

The ratio of C2/total dicarboxylic acids can be used to assess the aging process of organic aerosols, because oxalic acid has been recognized as the end product that is associated with atmospheric chain reactions of organic species with oxidants. Typically, higher ratios are observed with the progress of aerosol aging (Kawamura and Sakaguchi, 1999). Over south Asia, Indo-Gangetic plain is considered as a densely populated region, and thus as a potentially strong source region of anthropogenic aerosols. Northern part of these highly populated and industrialized areas is one of chains of Himalaya Mountains. Due to its high elevation, the Himalayan range acts as a boundary limiting the northern extent of the Indian summer monsoon, and therefore, observations at a high altitude location, Nainital (29.4° N; 79.5° E, 1958 m a.s.l.) would provide information about emissions over the Indian subcontinent. Their observation site is located at the highest mountain top (over Kumaon region) and about 2 km far from Nainital city (population ~0.5 million). The site is devoid of any major local pollution sources nearby and is generally free from the snow coverage during most of the time. North and northeast side of the study area are characterized by sharply peaking topography of Himalayan mountain ranges, whereas south-western side plains with very low elevation (<500 m a.s.l.) are densely populated with land merging into the Ganga basin (Hegde and Kawamura, 2012).

The C2/total diacid ratios show higher values in winter (~0.8±0.04) than summer (~0.5±0.01), suggesting that the winter aerosols may be more aged in Central Himalayas. As the anthropogenic aerosols that are emitted from the industrial regions of Indo

Gangetic Plain areas can travel to the north and reach the sampling site by the northerly wind (comparatively lower temperature and weaker wind speed) during the winter period, aging of these aerosols might occur during the transport and thereby significantly contribute to the higher C2/total diacid ratios. In contrast, this trend is reversed during summer. Because the temperature over the region increase and high wind favors quick transport of pollutants, fresher aerosols are transported over the sampling site (Hegde and Kawamura, 2012).

Please see line 333–338 in the revised MS. So it's rational to compare the values of C2/total diacid ratio in Beijing with those in Central Himalayas to find out that the photochemical formation of dicarboxylic acids is insignificant in urban Beijing.

(3) P13, L345-354, the discussion in this paragraph is hard to follow. Simply base on the relationships among these species and drawing this conclusion (Line 353-354) is not convincing.

Response: "$\omega C2$ and Pyr are more abundant in cold seasons (Table 1) with similar seasonal patterns (Fig. 3g–h). Both correlated well with K+ (Fig. S2) and Cl– (Fig. S3) in sampling seasons. These connections demonstrate that $\omega C2$ and Pyr originated from common combustion emissions or similar secondary formation pathways."

"Glyoxal (Gly) and methylglyoxal (MeGly) correlated well with nss-K+ (Gly: $0.3 \leq r2 \leq 0.9$, MeGly: $0.3 \leq r2 \leq 0.9$) throughout the whole year (Fig. S2), whereas Gly and MeGly showed good relations with Cl– (Gly: $0.3 \leq r2 \leq 0.8$, $0.4 \leq r2 \leq 0.8$) in autumn, winter and summer (Fig. S3). Concentrations of these two carbonyls are largely affected by biogenic precursors (e.g., isoprene and monoterpenes) emitted from vegetation and biomass burning activities during entire sampling periods in addition to coal burning and motor exhaust (aromatic hydrocarbons)." Please see line 297–300 and 315–321 in the revised MS.

Please see line 348–356 in the revised MS, where elaborated the meaning of correlation between C2/Pyr, C2/$\omega$C2, C2/Gly and C2/Tot.

No strong positive correlation was observed between $C_2$/Pyr, $C_2$/$\omega$C2, $C_2$/Gly and C2/Tot suggests that supplement of Pyr, $\omega$C2 and Gly were faster than their secondary

transformations to C2 in Beijing. The aging level was not strong enough for intermediate diacids contributing to the production of C2.

(4) P15, L405, The Ph/C6 ratio was lower than 2.5 in other seasons and it is only 4.1 in winter, but why it is concluded that more emissions from diesel burning (ratio of 6.58) than gasoline fuel vehicles (2.05)?

Response: We have rephrased the sentence. "This phenomenon shows abundances of diacids attributable to more emissions from gasoline fuel vehicles than diesel burning." Please see line 418–420 in the revised MS.

7, P20, L454, It is not clear what is connection between the information in L545-547 and the statement in L547-549.

Response: A previous study noted that increasing concentrations of oxalic and malonic acids inhibit the growth of total fungi number due to the lower pH, which in turn changes the efficiency of fungi to degrade the malonic acid (Côté et al., 2008).

Pavuluri and Kawamura (2012) found that $\delta 13C$ of LMW diacids (i.e., oxalic, malonic, succinic acids) in aged aerosols are less negative (i.e., enriched in 13C with an increase in WSOC/OC ratios), perhaps due to kinetic isotopic fractionation caused by ambient photo-chemical degradation of oxalic acid and also malonic acid. The efficiency of fungi can result in the degradation level of oxalic and malonic acids.

8, It is suggested to use the words of "significant" or "insignificant" in the statements carefully unless statistical data or solid evidence are provided.

Response: Thanks for your suggestion. We have corrected in some parts.

9, As presented in the manuscript, if I understood it correctly, the authors concluded that primary emissions are the major contributors to the organic acids and carbonyls. It is also concluded that the photochemical formation of these species in Beijing is insignificant. Both of these two statements are very strong. Is there any source apportionment study in Beijing during the same period, are they consistent or controversial?

Sorry, no similar study on the source apportionment was found in Beijing during the same period. Our paper concluded that primary emissions are the major contributors to the organic acids in Beijing, although photochemical formation of these species slightly enhanced in summer. Ji et al. (2016) observed increasing photochemical activity of aerosol in autumn and winter due to stagnant atmospheric conditions and inversion layer, which are favorable for the absorption and condensation of semi-volatile organic compounds on existing particles. And he noted that biomass burning is a substantial pollution factor throughout the year in Beijing. Different sampling time is related with various atmospheric conditions and strength of primary emissions. The sampling time in our study is not very special. Many scholars like to study the formation mechanisms and source apportionments of aerosols in haze days, while we focused more on the characterization of organic acids in common days including polluted and clean days in Beijing at a molecular level. Different sampling time and study target may get different conclusions.

Zhang et al. (2013) found that soil dust, coal combustion, biomass burning, traffic and waste incineration emission, industrial pollution, and secondary inorganic aerosol (SIA) are six main sources of $PM_{2.5}$ in Beijing. Each of these sources has an annual mean contribution of 16, 14, 13, 3, 28, and 26 %, respectively, to $PM_{2.5}$. Similarly, the relative contributions of these sources to $PM_{2.5}$ in Beijing greatly varied with the changing seasons, which proves the "complex air pollution" in Beijing. The highest contributions occurred in spring for soil dust (23 %) and traffic and waste incineration emission (5 %), in winter for coal combustion (57 %), in autumn for industrial pollution (42 %), in summer for SIA (54 %), and in both spring and autumn for biomass burning (19 and 17 %, respectively). The conclusions in this article demonstrated that regional primary sources could be crucial contributors to PM pollution in Beijing, compared to the enhanced photochemical formations of SIA in summer.

10, Figure 6, please describe the meanings of different symbols and percentages for the box. It is suggested to use same scale for different seasons.

Response: The y-axis in figure 6 showed the $\delta^{13}C$ values of diacids, glyoxylic and

pyruvic acids, and the x-axis represented the abbreviations of species detected in this study. Please see table 1 in the revised MS.

And we have redrawn the figure 6 with same scale.

Hegde, P., and Kawamura, K.: Seasonal variations of water-soluble organic carbon, dicarboxylic acids, ketocarboxylic acids, and α-dicarbonyls in Central Himalayan aerosols, Atmospheric Chemistry and Physics, 12, 6645-6665, 2012.

Ji, D., Zhang, J., He, J., Wang, X., Pang, B., Liu, Z., Wang, L., and Wang, Y.: Characteristics of atmospheric organic and elemental carbon aerosols in urban Beijing, China, Atmospheric Environment, 125, 293-306, 10.1016/j.atmosenv.2015.11.020, 2016.

Kawamura, K., and Sakaguchi, F.: Molecular distributions of water soluble dicarboxylic acids in marine aerosols over the Pacific Ocean including tropics, Journal of Geophysical Research: Atmospheres, 104, 3501-3509, 1999.

Liu, Y., Tao, S., Yang, Y., Dou, H., Yang, Y., and Coveney, R. M.: Inhalation exposure of traffic police officers to polycyclic aromatic hydrocarbons (PAHs) during the winter in Beijing, China, Science of the total environment, 383, 98-105, 2007.

Pavuluri, C. M., and Kawamura, K.: Evidence for 13-carbon enrichment in oxalic acid via iron catalyzed photolysis in aqueous phase, Geophysical Research Letters, 39, 3802, 2012.

Zhang, R., Jing, J., Tao, J., and Hsu, S. C.: Chemical characterization and source apportionment of PM2.5 in Beijing: seasonal perspective, Atmospheric Chemistry & Physics, 13, 7053-7074, 2013.

---

## Author Comment (AC3) · 2 Nov 2017

ACP review by Anonymous referee 3:

MS title: Molecular distribution and compound-specific stable carbon isotopic composition of dicarboxylic acids, oxocarboxylic acids, and α-dicarbonyls in PM2.5 from Beijing, China

Organic aerosols (OAs) account for major fraction of atmospheric particulate matter and also ubiquitous in nature. Among the OAs, the dicarboxylic acid and related polar compounds are one such widely studied chemical species that provide useful information about the relative significance of anthropogenic versus natural source contributions as well as primary emissions vs. secondary formation processes. In this context, combining the molecular distributions, concentrations, diagnostic mass ratios, air mass back trajectories and their stable carbon isotopic composition from this kind of studies are helpful in improving our current understanding of the complex nature of OAs. Therefore, the study is most relevant and publishable in ACP after a major revision.

I feel that conclusions are more clear and focused than the most of the text part of this MS. The comparison of mass ratios among seasons are too vague. This should be supported by the statistical analysis such as ANOVA. To me, comparison of mean and sd of mass ratios of dicarboxylic acids among seasons appear to be insignificant for this study. In order to truly appreciate the relative significance of various source emissions (biogenic vs. anthropogenic) based on mass ratios of dicarboxylic acids and other related polar compounds, I strongly recommend the authors to evaluate their seasonal datasets using a statistical test (e.g, ANOVA). I see that there is a missing link in terms of attributing the stable carbon isotopic composition of dicarboxylic acids' with source contributions. For example, how the lowest δ13C value of terephthalic acid (~-33.5‰) in winter indicates that it is emitted from plastic waste burning. Why not in other seasons? Is the plastic waste burning over Beijing is common only winter?

Response: Terephthalic acid was found to account for more than 77% of total diacids determined in aerosols from open-waste burnings (Kumar et al., 2015), and was a good tracer for fresh smoke waste burning particles (Kumar et al., 2015;Simoneit et al., 2005).

China has large rural population living in the village and straws are not a high-demand fuel. Hence during and after the harvest season, farmers often burn crop straws in the field as a convenient and inexpensive way to dispose agricultural waste to advance crop rotation (Fu et al., 2012;Wang et al., 2009;Ji et al., 2016). In most of the reports, open-waste burning are clubbed with biomass and fuel burning aerosols (Akagi et al., 2011;Lei et al., 2012). Substantial combustions activities, including fossil fuel burnings for domestic heating, open-waste and biomass burnings, occurred in autumn and winter. Furthermore, the inversion layer and low wind caused the accumulation of pollutants in winter, while low temperature reduced the photochemical processing of aerosols in Beijing. Thus, combining the conclusion found in previous studies, the comparison with other papers (figure 5), concentrations and $\delta^{13}C$ values of terephthalic acid (~-33.5‰), we think that abundant tPh is directly emitted by plastic waste burning in Beijing, especially for winter.

We have used principal component analysis (PCA) to discriminate the source apportionment of atmospheric aerosols. Please see section 3.5.

Another important issue is Section 3.6, 3.7 and 3.8: comparison of δ13C of diacids and other compounds measured here makes this study unique due to year round sampling and comparing seasons. However, all these sections are bit complicated to follow/read. Since the East Asian outflow influences the chemical composition of organic aerosols during winter and spring, it is relevant to compare the diacid δ13C values from this study with other sites/studies during this period only. No need to include autumn and summer. Therefore, I suggest authors to combine the winter and spring data sets and use the median values to compare with the other sites in E. Asia (e.g. Sapporo, Gosan & cruises).

Response: Thanks for your careful reading. Only combining the winter and spring data sets and using the median values to compare with the other sites may not show the year-round variation trend of $\delta^{13}C$ values and seasonal photochemical aging level of water-soluble organic acids in Beijing.

My other comments are as follows:

Line 35-37: The sentence is difficult to follow. Please rewrite.

Response: Rephrased. Thanks.

Line 38-40: Correlations of some oxocarboxylic acids and a-dicarbonyls with nss-K+, how significant these are? Mention clearly what is correlated with what? Some oxocarboxylic acids are not specific!

Response: Please see Page11, Line 297–300 and Page12, Line 315–321 in the revised MS, as well as figure S2&S3 in the supporting material. Major oxocarboxylic acids are glyoxylic and pyruvic acids, because glyoxylic acid is measured as the most abundant oxoacid, followed by pyruvic acid.

Line 188-190: What is the reason that oxalic acid concentration is found to be highest in autumn and lowest in winter? Similarly, What causes the difference in the seasonality for the relative abundances of oxalic acid? Why is it maximum in summer? Explain/suggest.

Response: Most organic compounds showed the highest concentrations in autumn and lowest values in winter, which are associated with the sampling time. Different sampling time is related with various atmospheric conditions and strength of primary emissions.

We conducted air mass back trajectories at the sampling location, which indicates most of the wind came from the southern industrialized areas and mixed with regional wind in autumn, spring and summer. From the figure 1 in the revised MS, we can see intensive biomass burnings in surroundings areas, especially for autumn and spring. Please see figure 3 in the revised MS. A big pollution process was observed from 10.3 to 10.8 in autumn, followed by secondary oxidation ways, which leaded to the increased concentrations of oxalic acid in fall. In contrast, the less polluted air parcels came from the northern areas during wintertime in Beijing. Although fossil fuel burning as a major source contributed to the abundance of diacids and related compounds like Ph, tPh, $\omega C_2$ and Pyr, the photodegradation of these organic acids to form oxalic acid was weak. Therefore, oxalic acid showed the lowest concentrations in winter.

Strong solar radiation and high temperature can accelerate the photochemical processing of aerosols. Typically, the prolonged photochemical oxidation of organics in the

atmosphere leads to enhanced relative abundance of oxalic acid in total diacids in summer.

Line 192: Update the references with a recent review.

Response: Updated. Please see Page 8, Line 197.

Line 195 to 197. Authors need to provide, why there exist differences in the molecular distribution of measured water-soluble organic compounds among seasons and why do they show different patterns for e.g., why the third most abundant compound is glyoxylic acid in cold period and malonic acid in warmer period?.

Response: Different molecular distributions of diacids and related compounds are attributed to strength of primary emissions and photochemical aging level, as well as atmospheric conditions. Please see the review published in 2016, which gave detailed information in various sampling seasons and areas (Kawamura and Bikkina, 2016).

Primary emissions, such as biomass burning, fossil fuel burning and motor vehicles, accounted for a large contribution of glyoxylic acid and related precursors. We conducted air mass back trajectories at the sampling location, which indicates most of the wind came from the southern industrialized areas and mixed with regional wind in autumn, spring and summer. From the figure 1 in the revised MS, we can see intensive biomass burnings in surroundings areas, especially for autumn and spring. Please see figure 3 in the revised MS. A big pollution process was observed from 10.3 to 10.8 in autumn, followed by secondary oxidation ways, which leaded to the increased concentrations of glyoxylic acid.

Meanwhile, the photochemical aging level of organic aerosols is weak in comparison with the strength of primary emissions in Beijing, although it enhanced in summer. These conclusions illustrate that the supplement of $\omega C_2$ is faster than its photodegradation to form other diacids in air. Low wind speed and inversion layer in autumn and winter can make species accumulate quickly, which makes $\omega C_2$ as the third most abundant compound in cold seasons.

Breakdown of relatively long carbon-chain diacids and other related precursors is one of

the key sources of low carbon-numbered diacids in the atmosphere. The photochemical contribution of related precursors to diacids, like the transformation of succinic acid to malonic acid, enhanced in warmer seasons due to increased relative humidity and temperature as well as stronger solar radiation. Thus malonic acid is the third most abundant compound malonic acid in warmer period.

Line 199-202: The sentence is not clear. What is single dicarboxylic acid?

Response: We have changed "single" to "individual". Please see Page 8, Line 204 in the revised MS.

Line 203: I am confused with subheading seasonal variability. Authors have already mentioned about differences in the molecular distributions of dicarboxylic acid among seasons already in the previous section. This section has to combine with the section 3.1.

Response: Thanks for your suggestion. We think it is better to divide this part into two sections. The section 3.1 is just to elaborate the seasonal molecular distributions, while the section 3.2 is to show the seasonal concentrations of water-soluble organic acids to discuss the possible sources in detail.

Line 209: abbreviate C9

Response: Corrected. Please see Page 9, Line 214 in the revised MS.

Line 204-213: The seasonal trends were attributed to different emissions. This is not enough. Explain what source emissions might contribute for each type and also justify why you think this is the only possibility?

Response: Please see following sentences in section 3.2–3.4 in the revised MS.

We have discussed in detail.

Line 214: Why the total diacid concentrations are the highest in autumn and why it is lowest in winter? Explain.

Response: Different sampling time is related with various strength of primary emissions

and atmospheric conditions. Please see figure 3 in the revised MS. A big pollution process was observed from 10.3 to 10.8 in autumn, followed by secondary oxidation ways, which leaded to the increased concentrations of organic acids, especially for C2–C4 diacids. These three diacids account for a large proportion of total diacids, thus total diacids had the highest concentrations in autumn.

In contrast, although there was a big pollution process from 12.20 to 12.26 in winter, it contributed much to the enhancement of Ph and tPh. Only small growth of concentrations was observed for $C_2$, $C_3$ and $C_4$ diacids due to insignificant secondary formation during wintertime, so concentrations of total diacids showed the lowest values in autumn.

Line 216: Why Beijing dataset has to be compared with Tanzania, Africa? Both are different settings? Compare with polluted atmosphere with another city in S. Asia or E. Asia. Given the diverse geographical locations, comparison with only one or two sites cannot be acceptable. Please compare or provide a table and discuss how different or similar this study site with those documented from other cities in China and India.

Response: We have also compared Beijing dataset with those in Tokyo, Japan and Gosan, Jeju island in Korea. Please see Page 9, Line 222–225 in the revised MS.

Please see table 2 in the revised MS. We discussed the differences and similarities of datasets between Beijing and other areas in all sections.

Line 223-230: After discussing the sources of oxalic acid, why there is a sudden jump to malonic acid data from this study. What about oxalic acid? If it is not important why authors are describing so much about its sources here. Connect here with their formation in different seasons. What are the different sources of oxalic acid, causing this variability through sampling period?

Response: Oxalic acid has been recognized as the end product that is associated with atmospheric chain reactions of organic species with oxidants. Malonic and succinic acids are important precursors of $C_2$. These diacids always have similar primary emissions or secondary formation ways. Please see Page 9, Line 240–242. In order to avoid repetition and link up with other parts, we have discussed its sources in detail in section 3.3.

Line 231: Why malonic acid is highest in autumn?

Response: Please see figure 3 in the revised MS. A big pollution process was observed from 10.3 to 10.8 in autumn, followed by secondary oxidation ways, which leaded to the increased concentrations of malonic acid.

Line 231-234: The connectivity between lines or sentences is missing. Why suddenly succinic acid to malonic acid ratio after mentioning the seasonal variability of malonic acid? What about the seasonal variability of succinic acid? Instead of picking up each compound measured and discussing its seasonality, I suggest authors to briefly summarize or infer logically the possible formation pathways of observed abundant compounds.

Response: Please see Page 9, Line 236–242 in the revised MS.

Line 231-235: Authors attributed the relative dominance of succinic acid over malonic acid as the major contribution from primary emissions to dicarboxylic acids measured here. Although this could be possible, however, one cannot rule about the transport during each season. So if you see the air mass back trajectories at the receptor site, then this inference based on C4/C3 has certain uncertainty or bias. So you need to mention this in the MS.

Response: We conducted air mass back trajectories at the sampling location, which indicates most of the wind came from the southern industrialized areas and mixed with regional wind in autumn, spring and summer. Please see figure 1 in the revised MS. Anthropogenic and biogenic organic compounds could have been mixed in some samples, followed by photochemical oxidation pathways. In contrast, clean north wind dominated in winter, but local inversion layer often occurred in Beijing, which caused the accumulation of organic aerosols. Because of the low temperature and weak solar radiation, the secondary oxidation level of aerosols may be not as strong as those in other seasons. Thus C3/C4 ratio showed the lowest mean value in winter.

C3 diacid can be produced as a result of hydrogen abstracted by OH radicals, followed by

decarboxylation processing of C4 diacid (Kawamura and Sakaguchi, 1999). The mass concentration ratio of C3/C4 is a good indicator for evaluating the contributions of dicarboxylic acids from primary emissions or secondary oxidation production in the atmosphere. The value of C3/C4 ratio is greater than unity for photochemically aged aerosols whereas it reaches 0.5 for vehicular emissions (Kawamura and Kaplan, 1987).

Line 235: The diurnal variation tendency of C2?? Is it diurnal or daily variability?

Response: Corrected. It's daily variability.

Line 255-257: Why the correlation of azelaic acid (C9) with K+ and Cl- solely attributed to coal burning? Why not biomass burning?

Response: "Azelaic acid correlated well with K+ ($0.3 \leq r2 \leq 0.4$) and Cl– ($0.4 \leq r2 \leq 0.5$) in cold seasons (Fig. S1), indicating that substantial amounts of C9 may be stemmed from the local and surrounding combustion activities in Beijing." Combustion activities include biomass/biofuel burnings and fossil fuel burnings.

Line 269-271: According to authors "The predominance of terephthalic acid over phthalic acid observed in this study is in contrast with those reported by Ho et al., ". Is it due to variability in the sources or increase plastic waste burning is increasing. Comment on this.

Response: It is due to the increase of plastic waste burning. Terephthalic acid is the tracer of plastic waste incineration. Please see Page 11, Line 277–279 in the revised MS.

Line 277-278: Provide a reference for the argument that monocarboxylic acids are photochemically oxidized & form dicarboxylic acid. Why authors think it is relevant here rather than direct emissions or other sources.

Response: We have added two references. Please see Page 11, Line 284 in the revised MS. All diacids, oxoacids and α-dicarbonyls can be emitted from primary emissions or formed via photochemical oxidation of related precursors in atmosphere. We have discussed their direct emissions and secondary photodegradations in line 294–305 and

355–367.

Line 281: the sentence is not clear.

Response: Corrected. Please see line 287–289 in the revised MS.

Line 290-292: These sentences are not clear. You can combine into one as "wC2 and Pyr is more abundant in cold seasons (Table 1) and correlated with K+ and Cl-". What is the common combustion source, mention it?

Response: Potassium ion (K+) is a good tracer of biomass burnings (Andreae, 1983). And a great deal of chloride in wintertime Beijing is linked to increased emissions of coal incineration, particularly under stagnant meteorological conditions that facilitate the formation of particle-phase ammonium chloride (Sun et al., 2013). $\omega$C2 and Pyr show similar seasonal patterns (Fig. 3g–h) and correlated well with K+ (Fig. S2) and Cl– (Fig. S3) in cold seasons, which demonstrate that $\omega$C2 and Pyr originated from common combustion emissions like biomass burnings, fossil fuel burnings. We have rephrased. Please see line 297–300 in the revised MS.

Line 326-328: I don't follow the comparison with the Central Himalayan aerosols as well as the logic of the statement. rewrite.

Response: The ratio of C2/total dicarboxylic acids can be used to assess the aging process of organic aerosols, because oxalic acid has been recognized as the end product that is associated with atmospheric chain reactions of organic species with oxidants. Typically, higher ratios are observed with the progress of aerosol aging (Kawamura and Sakaguchi, 1999).

Over south Asia, Indo-Gangetic plain is considered as a densely populated region, and thus as a potentially strong source region of anthropogenic aerosols. Northern part of these highly populated and industrialized areas is one of chains of Himalaya Mountains. Due to its high elevation, the Himalayan range acts as a boundary limiting the northern extent of the Indian summer monsoon, and therefore, observations at a high altitude location, Nainital (29.4° N; 79.5° E, 1958 m a.s.l.) would provide information about

emissions over the Indian subcontinent. Their observation site is located at the highest mountain top (over Kumaon region) and about 2 km far from Nainital city (population ~0.5 million). The site is devoid of any major local pollution sources nearby and is generally free from the snow coverage during most of the time. North and northeast side of the study area are characterized by sharply peaking topography of Himalayan mountain ranges, whereas south-western side plains with very low elevation (<500 m a.s.l.) are densely populated with land merging into the Ganga basin (Hegde and Kawamura, 2012).

The C2/total diacid ratios show higher values in winter (~0.8±0.04) than summer (~0.5±0.01), suggesting that the winter aerosols may be more aged in Central Himalayas. As the anthropogenic aerosols that are emitted from the industrial regions of Indo Gangetic Plain areas can travel to the north and reach the sampling site by the northerly wind (comparatively lower temperature and weaker wind speed) during the winter period, aging of these aerosols might occur during the transport and thereby significantly contribute to the higher C2/total diacid ratios. In contrast, this trend is reversed during summer. Because the temperature over the region increase and high wind favors quick transport of pollutants, fresher aerosols are transported over the sampling site (Hegde and Kawamura, 2012).

The C2/Tot ratios were the lowest in winter (0.39±0.05), while C2/Tot ratios are similar in the other three seasons (0.54–0.58). So it's rational to compare the values of C2/total diacid ratio in Beijing with those in Central Himalayas to find out that the photochemical formation of dicarboxylic acids is insignificant in urban Beijing.

We have rephrased the sentences. Please see line 333–338 in the revised MS.

Line 333-339: In urban Beijing, how can authors assume that succinic acid formation forms the photooxidation of unsaturated fatty acids? What about the photochemical oxidation of adipic acid, which is a product of cyclic olefins with oxidants in and around the city? Why not is C4 derived from anthropogenic emissions such as fossil fuel combustion, vehicular emissions in Beijing? The linear relationship between C2/Tot (or relative abundance of oxalic acid in total diacid mass) and the C2/C4 just indicate that

oxalic acid has a significant contribution from or formed from the photochemical oxidation of succinic acid, not more than that. So authors need to dilute their emphasis on source attribution directly based on a linear relationship. If still, authors think that succinic acid might have produced from the photochemical breakdown of higher homologues of dicarboxylic acids from the biogenic unsaturated fatty acids, they should the linear relationships with oleic acid and azelaic acid first and then lower homologues of dicarboxylic acids with azelaic acid. I suggest authors think and rewrite along these lines.

Response: In figure 3, a big pollution process was observed from 10.3 to 10.8 in autumn, followed by secondary oxidation ways, which leaded to the increased concentrations of organic acids in fall. C2, C4, C9, ωC2 and Pyr showed similar concentration variations, indicating they may have common primary sources or photochemical pathways. In section 3.2, C9, ωC2, Pyr, Gly and MeGly correlated well with nss-K+, further implying that these organic acids were emitted from biomass burnings.

We have rephrased. "Typically, concentrations of C2, C4 and C9 diacids, as well as ωC2 and Pyr showed similar varational trends in autumn, implying that they were derived from similar primary emissions or photochemical processing. Furthermore, C2/Tot (C2%) showed strong correlations with C2/C4 in all four seasons (Fig. S4), indicating the importance of biogenic unsaturated fatty acids, followed by photochemical processing."
Please see line 344–348 in the revised MS.

Line 335: Authors need to provide a proper reference for invoking contribution of emissions from the phytoplankton in remote oceans (update the reference with diacid review, doi: 10.1016/j.atmosres.2015.11.018 and others cited in).

Response: We have updated the reference with the diacid review published in 2016. Please see Page 13, Line 343 in the revised MS.

Line 340-344 and 345-346 is missing. Connect these two as "We, therefore, would like to investigate the significance of these formation pathways by examining the interrelationships between..."

Response: Thanks for your suggestions. Please see line 355–357 in the revised MS.

Line 347: correct the sentence as "in all seasons except winter"
Response: Corrected. Please see Page 14, Line 358.

Line 352-353: how the negative correlation between oxalic to glyoxylic acid mass ratio and the relative abundance of oxalic acid in summer demonstrate that C2, wC2, Pyr and Gly are from biomass burning emissions? Elaborate further.
Response: 1. Pyr, $\omega C_2$, Gly and MeGly correlated well with K+ in the sampling time, indicating that biomass burning contributed a large part of these organic acids.
2. Please see line 348–357 in the revised MS, where elaborated the meaning of correlation between C2/Pyr, C2/$\omega$C2, C2/Gly and C2/Tot.
3. No strong positive correlation was observed between $C_2$/Pyr, $C_2$/$\omega$C2, $C_2$/Gly and C2/Tot suggests that supplement of Pyr, $\omega$C2 and Gly were faster than their secondary transformations to C2 in Beijing. The aging level was not strong enough for intermediate diacids contributing to the production of C2.
4. Combining the results obtained in section 3.2 and relationships between C2/Tot and C2/Pyr, C2/$\omega$C2, C2/Gly, we found that C2, Pyr, $\omega$C2 and Gly are from biomass burning emissions.

Line 361: provide a reference that C3/C4 is a good indicator for dicarboxylic acid contribution from primary emissions vs. secondary formation process.
Response: Provided. Please see Page 14, Line 372.

Line 366: correct the sentence as "throughout the sampling year"
Response: Corrected. Please see Page 14, Line 377.

Line 367-368. I couldn't find any difference between the two statements.
Response: Thanks for your careful reading. We have rephrased. Please see Page 14, Line 377–379.

Line 376-377: Confusing!. State clearly that the direct emissions from localized sources in Beijing contributed to atmospheric dicarboxylic acids.

Response: "These results demonstrated that in addition to slightly enhanced atmospheric photochemical reactions in summer, incomplete combustions, like motor vehicles and biomass burnings, overwhelmingly contributed to dicarboxylic acids in Beijing." We have rephrased. Please see line 385–387 in the revised MS.

Line 380-384: I understand for the summer season that Ph might have a contribution from photochemical oxidation of PAHs. What is the reason for its higher abundance in winter season?

Response: Phthalic acid is either formed via photochemical pathways of naphthalene, or directly released into air by fossil fuel burning and the incomplete combustion of aromatic hydrocarbons in motor vehicles. Moreover, the abundance of Ph may also be caused by increased phthalates emissions from plastic waste burnings in heavily polluted areas in China (Deshmukh et al., 2015). Phthalic acid esters are used as plasticizers in resins and polymers (Simoneit et al., 2005). The Ph/Tot ratios in winter were nearly 2–3 times greater than those in spring and autumn, which was caused by substantial coal burning for house heating in Beijing. Therefore, these findings imply that phthalic acid is largely emitted by anthropogenic sources in winter, mainly as a result of intensive fossil fuel combustion.

Line 396-398: As stated in Line 385, if both adipic acid (C6) and phthalic acid (Ph) are produced from the photochemical oxidation of cyclic olefins, why C6/C9 is lowest in winter and whereas Ph/C9 is highest in winter. Perhaps both could have been produced by different sources. That is the reason why comparing seasonal means directly can yield erroneous interpretations. May be the seasonal averages are not significantly different.

Response: C6 is formed via secondary oxidations of cyclic olefins (e.g., cyclohexene), and Ph is a photochemical product of aromatic hydrocarbons. The precursor sources of C6 and Ph are technically defined as anthropogenic emissions. In contrast, C9 is mainly

produced by photochemical oxidation of biogenic unsaturated fatty acids. Thus, the mass concentration ratios of C6/C9 and Ph/C9 may effectively indicate the source strength of anthropogenic and biogenic emissions to these organic acids.

Due to large emission of fossil fuel burning, Ph showed the highest concentrations in winter. Moreover, the abundant presence of Ph may also be caused by enhanced emission of phthalates from plastics used in heavily populated and industrialized regions in China. Phthalic acid esters are used as plasticizers in resins and polymers. Thus Ph/C9 is highest in winter.

Previous study reported that contributions of C6 and Ph from biomass burning are minimal (Kawamura et al., 2013). Biomass burning is a substantial polluted factor in Beijing. Compared to the emission strength of biomass burning, the source of cyclic olefins may be not strong. Therefore, mean values of C6/Tot are constantly low in all four seasons, whereas the seasonal ratios of C9/Tot are the highest (0.09) in winter due to stagnant atmospheric conditions, which result in the lowest value of C6/C9 ratios in winter (0.34±0.13).

Line 410-414: I am not able to follow the comparison of ratios. Authors need to evaluate the differences in the seasonal means using ANOVA.

Why not authors use PMF other than PCA for the source apportionment in this study, like their previous publications?

Response: Being similar to C3/C4 ratio, mass ratios of M/F provide some clues about aging of air masses because lower values are caused by photochemical isomerization from M (cis) to F (trans), representative of higher photochemical activities or viceversa (Kawamura and Bikkina, 2016).

Previous publications always use PCA for the source apportionment, because PMF need a large quantity of data to run in order to get a accurate source analysis. But the data of diacids and related compounds are not enough for PMF.

Section 3,4: I do not understand the title of this subheading. Already authors made a comparison in the previous section. Then why suddenly there is another section on this?

Response: The section 3.3 is correlation analysis and seasonal concentration ratios of diacids for sources' discussion in Beijing, while the section 3.4 shows the comparisons of the mean mass ratios of organic acids in our study with those in other polluted areas. The conclusions in section 3.4 further prove the results discussed in previous sections.

Line 519-523: I do not follow the logic here. How authors drew their conclusion that d13C of C9 indicate that azelaic acid is from biomass burning in surrounding areas?

Response: It is important to state that marine-derived particulate organic matter in the remote marine atmosphere (around −20‰) is somewhat more enriched with 13C than those reported for terrestrial vegetation (C3 plants; −27‰) (Turekian et al., 2003).

In contrast, the smaller δ13C values of C6 (−25.8‰) and C9 (−28.1‰) in the Sapporo aerosols suggest more contribution from anthropogenic sources and terrestrial higher plants (biomass burning) (Aggarwal and Kawamura, 2008).

We have rephrased. Please see line 536–541 in the revised MS.

Line 534-535: How the lowest δ13C value of terephthalic acid (~-33.5‰) indicates that it is emitted from plastic waste burning that too in winter. Why not in other seasons? Is the plastic waste burning over Beijing is common only winter. Line 534-535 and the next sentence are not connected. The reference cited is the work related to S. Asia. How about the conditions in E. Asia? Is the plastic waste burning is relevant for the study site? If so, please state it.

Response: Terephthalic acid was found to account for more than 77% of total diacids determined in aerosols from open-waste burnings (Kumar et al., 2015), and was a good tracer for fresh smoke waste burning particles (Kumar et al., 2015;Simoneit et al., 2005). China has large rural population living in the village and straws are not a high-demand fuel. Hence during and after the harvest season, farmers often burn crop straws in the field as a convenient and inexpensive way to dispose agricultural waste to advance crop rotation (Fu et al., 2012;Wang et al., 2009;Ji et al., 2016). In most of the reports, open-waste burning are clubbed with biomass and fuel burning aerosols (Akagi et al., 2011;Lei et al., 2012). Substantial combustions activities, including fossil fuel burnings

for domestic heating, open-waste and biomass burnings, occurred in autumn and winter. Furthermore, the inversion layer and low wind caused the accumulation of pollutants in winter, while low temperature reduced the photochemical processing of aerosols in Beijing. Thus, combining the conclusion found in previous studies, the comparison with other papers (figure 5), concentration and $\delta^{13}C$ values of terephthalic acid (~-33.5‰), we think that abundant tPh is directly emitted by plastic waste burning in Beijing, especially for winter.

Line 538: What major compounds? Be specific (diacids, oxoacids or a-dicarbonyls or what you are referring to).

Response: Diacids includes oxalic, malonic, succinic, glutaric, adipic, azelaic, phthalic and terephthalic acids. The oxoacids includes glyoxylic and pyruvic acids.

Line 540: The decreasing trend in d13C of C5 to C9 is not obvious from the Figure 6 for all seasons (see for e.g., summer, panel d).

Response: Deleted.

Line 542-544: given the overlap of the box widths for oxalic, malonic and succinic acids, I wonder whether the 13C enrichment for these compounds is significantly different among seasons.

Response: The mean $\delta$13C values of C2 and C3 were constant among seasons. In contrast, succinic acid showed the lowest values (–28.6‰) in summer. Please see table 5 in the revised MS.

The scale in Figure 6 is most difficult to compare between seasons. I suggest authors keep the same scale and then discuss the differences.

Response: Thanks for your suggestion. We have redrawn.

Figure 8: I do not follow the caption. What are the concentrations plotted on the x-axis.

Response: These are the seasonal concentrations of C2 plotted on the x-axis.

References:

Aggarwal, S. G., and Kawamura, K.: Molecular distributions and stable carbon isotopic compositions of dicarboxylic acids and related compounds in aerosols from Sapporo, Japan: Implications for photochemical aging during long-range atmospheric transport, Journal of Geophysical Research, 113, 10.1029/2007jd009365, 2008.

Akagi, S. K., Yokelson, R. J., Wiedinmyer, C., and Alvarado, M. J.: Emission factors for open and domestic biomass burning for use in atmospheric models, Atmospheric Chemistry & Physics, 11, 27523-27602, 2011.

Andreae, M. O.: Soot carbon and excess fine potassium: long-range transport of combustion-derived aerosols, Science, 220, 1148, 1983.

Fu, P., Kawamura, K., Chen, J., Li, J., Sun, Y., Liu, Y., Tachibana, E., Aggarwal, S., Okuzawa, K., and Tanimoto, H.: Diurnal variations of organic molecular tracers and stable carbon isotopic composition in atmospheric aerosols over Mt. Tai in the North China Plain: an influence of biomass burning, Atmospheric Chemistry and Physics, 12, 8359-8375, 2012.

Hegde, P., and Kawamura, K.: Seasonal variations of water-soluble organic carbon, dicarboxylic acids, ketocarboxylic acids, and α-dicarbonyls in Central Himalayan aerosols, Atmospheric Chemistry and Physics, 12, 6645-6665, 2012.

Ji, D., Zhang, J., He, J., Wang, X., Pang, B., Liu, Z., Wang, L., and Wang, Y.: Characteristics of atmospheric organic and elemental carbon aerosols in urban Beijing, China, Atmospheric Environment, 125, 293-306, 10.1016/j.atmosenv.2015.11.020, 2016.

Kawamura, K., and Kaplan, I. R.: Motor exhaust emissions as a primary source for dicarboxylic acids in Los Angeles ambient air, Environmental Science & Technology, 21, 105-110, 1987.

Kawamura, K., and Sakaguchi, F.: Molecular distributions of water soluble dicarboxylic acids in marine aerosols over the Pacific Ocean including tropics, Journal of Geophysical Research: Atmospheres, 104, 3501-3509, 1999.

Kawamura, K., Tachibana, E., Okuzawa, K., Aggarwal, S., Kanaya, Y., and Wang, Z.: High abundances of water-soluble dicarboxylic acids, ketocarboxylic acids and

α-dicarbonyls in the mountaintop aerosols over the North China Plain during wheat burning season, Atmospheric chemistry and physics, 13, 8285-8302, 2013.

Kawamura, K., and Bikkina, S.: A review of dicarboxylic acids and related compounds in atmospheric aerosols: Molecular distributions, sources and transformation, Atmospheric Research, 170, 140-160, 2016.

Kumar, S., Aggarwal, S. G., Gupta, P. K., and Kawamura, K.: Investigation of the tracers for plastic-enriched waste burning aerosols, Atmospheric Environment, 108, 49-58, 2015.

Lei, W., Li, G., and Molina, L. T.: Modeling the impacts of biomass burning on air quality in and around Mexico City, Atmospheric Chemistry & Physics, 12, 2299-2319, 2012.

Simoneit, B. R., Medeiros, P. M., and Didyk, B. M.: Combustion products of plastics as indicators for refuse burning in the atmosphere, Environmental science & technology, 39, 6961-6970, 2005.

Sun, Y., Wang, Z., Fu, P., Yang, T., Jiang, Q., Dong, H., Li, J., and Jia, J.: Aerosol composition, sources and processes during wintertime in Beijing, China, Atmospheric Chemistry and Physics, 13, 4577-4592, 2013.

Turekian, V. C., Macko, S. A., and Keene, W. C.: Concentrations, isotopic compositions, and sources of size-resolved, particulate organic carbon and oxalate in near-surface marine air at Bermuda during spring, Journal of Geophysical Research, 108, 347-362, 2003.

Wang, G. H., Kawamura, K., Xie, M. J., Hu, S. Y., Cao, J. J., An, Z. S., Waston, J. G., and Chow, J. C.: Organic molecular compositions and size distributions of Chinese summer and autumn aerosols from Nanjing: characteristic haze event caused by wheat straw burning, Environmental Science & Technology, 43, 6493, 2009.

---

## Author Response (AR1)

2nd November 2017

Professor Hang Su

Editor of *Atmospheric Chemistry and Physics*

RE: acp-2017-410

Dear Prof. Su

Enclosed please find a revised manuscript #acp-2017-410, "Molecular distribution and compound-specific stable carbon isotopic composition of dicarboxylic acids, oxocarboxylic acids, and α-dicarbonyls in PM2.5 from Beijing, China" by Wanyu Zhao et al.

 Here, we have carefully revised the manuscript based on all of the comments and suggestions from three referees to assure the validity of the results and interpretation. Please find the details in the author's responses to the reviewers and in the revised manuscript. We believe that the revised manuscript is now ready to be accepted for publication in *Atmospheric Chemistry and Physics* as a research article. Thank you very much for your kind consideration!

Sincerely yours,

Pingqing Fu, Ph.D

Professor of Atmospheric Chemistry and Biogeochemistry

Institute of Atmospheric Physics, Chinese Academy of Sciences

Beijing 100029, China

e-mail: fupingqing@mail.iap.ac.cn

phone: 86-10-8201-3200